# Artificial Soils Reveal Individual Factor Controls on Microbial Processes

Ilenne Del Valle,[a]* Xiaodong Gao,[b] Teamrat A. Ghezzehei,[c] Jonathan J. Silberg,[d,e,f] Caroline A. Masiello[b,d,g]

[a]Systems, Synthetic, and Physical Biology Graduate Program, Rice University, Houston, Texas, USA
[b]Department of Earth, Environmental, and Planetary Sciences, Rice University, Houston, Texas, USA
[c]Department of Life and Environmental Sciences, University of California, Merced, California, USA
[d]Department of BioSciences, Rice University, Houston, Texas, USA
[e]Department of Bioengineering, Rice University, Houston, Texas, USA
[f]Department of Chemical and Biomolecular Engineering, Rice University, Houston, Texas, USA
[g]Department of Chemistry, Rice University, Houston, Texas, USA

**ABSTRACT** Soil matrix properties influence microbial behaviors that underlie nutrient cycling, greenhouse gas production, and soil formation. However, the dynamic and heterogeneous nature of soils makes it challenging to untangle the effects of different matrix properties on microbial behaviors. To address this challenge, we developed a tunable artificial soil recipe and used these materials to study the abiotic mechanisms driving soil microbial growth and communication. When we used standardized matrices with varying textures to culture gas-reporting biosensors, we found that a Gram-negative bacterium (*Escherichia coli*) grew best in synthetic silt soils, remaining active over a wide range of soil matric potentials, while a Gram-positive bacterium (*Bacillus subtilis*) preferred sandy soils, sporulating at low water potentials. Soil texture, mineralogy, and alkalinity all attenuated the bioavailability of an acyl-homoserine lactone (AHL) signaling molecule that controls community-level microbial behaviors. Texture controlled the timing of AHL sensing, while AHL bioavailability was decreased $\sim10^5$-fold by mineralogy and $\sim10^3$-fold by alkalinity. Finally, we built artificial soils with a range of complexities that converge on the properties of one Mollisol. As artificial soil complexity increased to more closely resemble the Mollisol, microbial behaviors approached those occurring in the natural soil, with the notable exception of organic matter.

**IMPORTANCE** Understanding environmental controls on soil microbes is difficult because many abiotic parameters vary simultaneously and uncontrollably when different natural soils are compared, preventing mechanistic determination of any individual soil parameter's effect on microbial behaviors. We describe how soil texture, mineralogy, pH, and organic matter content can be varied individually within artificial soils to study their effects on soil microbes. Using microbial biosensors that report by producing a rare indicator gas, we identify soil properties that control microbial growth and attenuate the bioavailability of a diffusible chemical used to control community-level behaviors. We find that artificial soils differentially affect signal bioavailability and the growth of Gram-negative (*Escherichia coli*) and Gram-positive (*Bacillus subtilis*) microbes. These artificial soils are useful for studying the mechanisms that underlie soil controls on microbial fitness, signaling, and gene transfer.

**KEYWORDS** acylhomoserine lactone, artificial soils, biosensor, indicator gas, cell signaling, soil, synthetic biology, water retention curve

Address correspondence to Caroline A. Masiello, masiello@rice.edu.

*Present address: Ilenne Del Valle, Biosciences Division, Oak Ridge National Laboratory, Oak Ridge, Tennessee, USA.

The authors declare no conflict of interest.

The soil biome is a critical component of planetary-scale biogeochemical processes (1–4). A wide range of soil properties influences the soil biome's growth, distribution, and metabolism (5, 6). While it is clear that we need to understand how microbial

behaviors underlying biogeochemical processes vary dynamically to predict their control of planetary-scale processes in response to climate change, we do not yet understand the mechanisms by which soils modulate microbial growth, interactions mediated by signaling molecules, and perception of nutrients in their environment.

Soils' extraordinary heterogeneity and complexity makes it challenging to develop mechanistic insight into soil controls on microbial behaviors. Individual soil properties, such as redox potential, pH, bioavailable nutrients, and water availability, can vary by orders of magnitude within and between soil aggregates (7, 8). Because of the large variation in soil properties, it is challenging to design standardized, reproducible experiments to understand which specific matrix properties act to control microbial behaviors. These circumstances raise the need for matrices more complex than liquid medium, which is typically used in synthetic biology studies, but simpler than natural soils.

Artificial matrices allow studies that examine how specific soil parameters affect microbial behaviors while keeping other parameters constant. These synthetic matrices are an extension of the CLORPT model (CL = climate, O = organisms, R = relief, P = parent material, and T = time) for soil formation and logical addition to our use of soil sequences (9–11). Artificial soils allow experiments where soil properties can be modified systematically, enabling the study of the effects of individual soil properties on the behavior of the soil system. Simplified matrices built in the past have yielded valuable results (12–15). For instance, beads made out of transparent polymers demonstrated the importance of a matrix in shaping root growth and structure and plant-microbe interactions (16, 17). However, these simple beads did not mimic the effect of soil chemistry and mineralogy on the soil-plant-microbe interactions.

Artificial soils have been built by mixing the primary components of soils, such as sand, clays, and humic acids (12, 18). Studies with these synthetic matrices have provided insight into how different starting materials affect soil formation (19–22). Additionally, they have revealed how soil properties influence microbial community composition (23–26) and enzyme activity (18). However, existing protocols for artificial soils do not offer customizable recipes to create simplified soils with full control on all properties. They also do not allow individual physical and chemical soil characteristics to be varied independently. This limitation makes it difficult to adapt these soil recipes for studies that seek to examine the effect of individual soil characteristics (particle size, mineralogy, percentage of organic matter, or aggregation) on microbial behaviors.

Here, we report a flexible protocol for producing artificial soils capable of acting as a standardized matrix, intermediate between the petri dish and natural soils. These artificial soils have a range of tunable features that can be independently varied, including particle size distribution, mineral composition, pH, and organic matter (OM) content and composition. Using this protocol, we created seven artificial soils and characterized their physical (water retention curve, surface area) and chemical (pH) properties. By combining the artificial soils with a new synthetic biology reporting tool, gas biosensors (27, 28), we establish how microbial growth and interorganism communication vary dynamically with soil physicochemical properties. Particularly, we use these artificial soils to explore the effect of individual soil properties on the bioavailability of an acylhomoserine lactone (AHL) signal that underlies many forms of microbe-microbe communication (29, 30). We find that all matrix properties decrease signal bioavailability albeit to differing extents.

## RESULTS

**Artificial soil production.** In our design, we sought to control critical inorganic and organic soil properties that could affect the soil system, specifically tailoring development to those properties relevant to biome behavior. We mimicked soil physical properties by varying grain size (sand, silt, clay) and aggregation, and soil chemical properties by changing minerals, pH, organic carbon, and nitrogen content. These properties together act to control gas flow and soil water properties like hydraulic conductivity and soil water energy

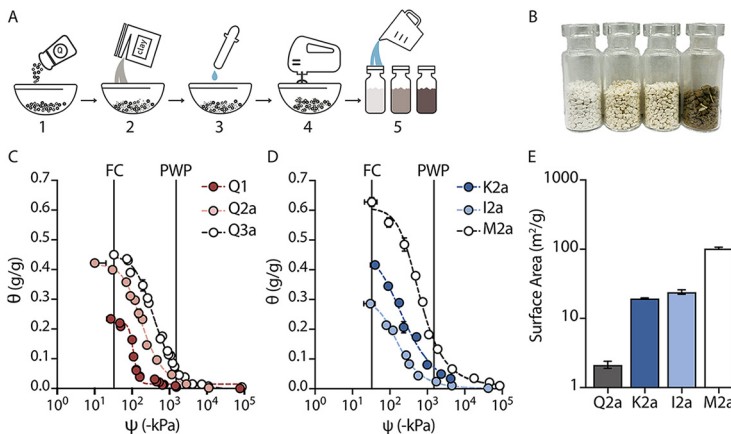

**FIG 1** Design and characterization of artificial soils. (A) Artificial soils produced by: (1) mixing quartz of different sizes together to provide texture; (2) adding clay minerals to vary mineralogy; (3) adjusting pH using $CaCO_3$; (4) aggregating using wet-dry cycles; and (5) hydrating to the desired water content ($\theta$) and potential ($\psi_m$). (B) Three artificial soils (Q2a, M2a, and Q2x0.5) and a natural soil. (C) Water retention curves of soils that vary only in texture or (D) mineralogy. Plant available water follows the trend Q2a > Q3a > Q1. (E) Surface area of soils that vary in mineralogy. Error bars represent one standard deviation from three experiments.

as characterized by a water retention curve (31–33). We judged pore connectivity, along with access to $O_2$, water, and nutrients, as central parameters for microbial life.

In the first step of soil construction, a nonreactive matrix of controlled particle size distribution is generated (Fig. 1A). This step is critical because grain size regulates soil microbial diversity by providing different microenvironments (pore spaces) and controlling the diffusivity of nutrients (34). To control particle size distribution, we mixed three diameters of monodisperse quartz ($SiO_2$), including sand ($\sim$70 $\mu$m), silt ($\sim$8.7 $\mu$m), and clay ($\sim$ 1.7 $\mu$m). This process allows the construction of soil textures that span the natural soil range (35). The distribution among the three particles sizes can be chosen by the user to mimic desired soil textures (Fig. S1A). For this study we chose three textures, including sand (90% sand, 5% silt, 5% clay) named Quartz-1 (Q1), silt loam (20% sand, 60% silt, 20% clay) named Quartz-2 (Q2), and clay (20% sand, 20% silt, 60% clay) named Quartz-3 (Q3). When aggregated, the names of each were modified (e.g., Q2a).

In the second production step reactive minerals are added. Soils contain a variety of minerals derived from natural weathering processes, which can carry positive or negative surface charges (31, 33). These charged surfaces contribute to nutrient ion holding capacity (36, 37). Minerals can also sorb organic compounds, and when dissolved can become a source of micronutrients for microbes (e.g., goethite [$\alpha$-FeOOH] can be an iron source) (38). Furthermore, minerals offer different surfaces for microbial interaction and colonization, and they can impact microbial survival and soil formation (23, 25, 26). Diverse reactive minerals can be added to the quartz matrix depending on the particular biome-mineral interaction to be explored. For this study, we use three common phyllosilicate clay minerals widely found in soils to create artificial silt loam soils, including kaolinite (K2a), illite (I2a), and montmorillonite (M2a). These minerals represent the major structural classes of soil clay minerals which are ubiquitous across natural soils (Fig. S1B).

In the third step the artificial soil pH is adjusted. Soil pH values typically vary from 5 to 8 (39). Soil pH is one of the primary determinants of total microbial biomass and community structure, with fungi tolerating a wider range of pH than bacteria (40–42). Soil pH also regulates microbial metabolism and communication (29, 43, 44), influencing the half-life of some signals used for cell-cell communication (29). The quartz matrices used here (Q1, Q2a, and Q3a) are at a neutral pH, but more basic soils can be generated through the addition of $CaCO_3$. More acidic soils can be created by adding aluminum

sulfate, or other sources of hydrogen ions. In this work, we explored the addition of $CaCO_3$ to increase the pH of a quartz-based clay soil (Q3a-pH 8) to a value of pH = 8 (Fig. S1C).

In the last step, aggregation is used to create structure so that artificial matrices more accurately mimic the fluid flow and gas diffusion of natural soils. In nature, aggregate formation is a complex process controlled by physical (sequential drying/wetting, freezing/thawing), chemical (electrostatic interactions, cation bridging), and biological (fungal hyphae, bacterial exopolysaccharides, and root exudates gluing particles together) processes (8). Aggregation is essential to simulate environmentally relevant soil water conditions, as large pore spaces created by aggregation are crucial to the maintenance of enough $O_2$ to support the aerobic soil biome in certain soil types, e.g., clay soils under saturated conditions (6, 45). Aggregation also creates specialized niches that allow the coexistence of aerobes and anaerobes in unsaturated soils (46, 47), allowing for more complex biogeochemical reactions to occur. Finally, the physical disconnection between aggregates influences evolutionary trajectories in isolated communities (48).

We explored two methods to simulate natural soil structure. In the first approach, we used wet-dry cycling to aggregate quartz particles through weak adhesion (12). With this approach, mineral addition leads to increased aggregate stability. Although these inorganic peds are fragile, they can survive autoclaving, providing structure in the absence of organic carbon (a ped is the fundamental unit of aggregation for a given soil). With artificial soils, it is beneficial to have the option to create aggregates lacking OM so that individual soil properties can be varied independently. In the second method, we added extracellular polymeric substances (EPS) prior to subjecting the matrix to wet/dry cycles (Fig. S1d). In this study we added xanthan and chitin to soils at 0.5 and 1% OM (wt/wt). This approach was used rather than allowing microbes to grow and form EPS in the matrix, since this latter approach takes months to generate EPS while our approach generates OM in a single day (49). Xanthan, a natural EPS produced by Gram-negative bacteria, has been used previously to represent soil EPS in model systems (50–55). Xanthan increases soil water holding capacity, aggregate stability, and tensile strength; it can also serve as a carbon source for microbes (50, 54). Chitin, a fungal- and insect-derived polymer containing carbon and nitrogen, has also been used as a model OM compound due to its abundance in soil ecosystems where it acts as a nutrient source (56, 57). Because not all microbes are able to use xanthan or chitin as nutrient sources, the addition of these OM sources allow for experiments that only test the physical effects of OM.

**Artificial soil characterization.** To determine how artificial soil composition relates to physicochemical properties, we characterized the water retention, surface area, and pH of each soil (Fig. 1B). Water properties were evaluated because they determine plant and microbe viability. Water content ($\theta$, g[$H_2O$]/g[soil]) alone is not a sufficient descriptor, because soils often hold the same amount of water at different water potentials. For example, at a single soil water content water may be biologically accessible in a sandy soil but not in a clay soil. To understand hydration conditions, it is most meaningful to measure water retention curves (WRC), which relate soil water content ($\theta$) and soil water potential ($\psi$). We generated WRC for sets of soils that differed in particle size distribution (Fig. 1C), mineralogy (Fig. 1D), and pH (Fig. S2A); the data were fit to the van Genuchten model to obtain hydraulic parameters (Table S1) (58, 59).

Plant available water (PAW), defined as the water held by any given soil between the potentials defined by field capacity (FC) at $\psi_m = -33$ kPa and permanent wilting point (PWP) at $\psi_m = -1500$ kPa, was highest in the silt loam (Q2a; PAW = 0.37 g[$H_2O$]/g[soil]) followed by clay (Q3a; PAW = 0.36 g[$H_2O$]/g[soil]) and sandy (Q1; PAW = 0.22 g[$H_2O$]/g[soil]) soils. Silt loam soils with a 1:1 clay (kaolinite; K2a) and an expanding 2:1 clay (montmorillonite; M2a) had higher PAWs with values corresponding to 0.42 g($H_2O$)/g(soil) and 0.60 g($H_2O$)/g(soil), respectively. The soil with a 2:1 nonexpanding clay (illite; I2a) had a PAW = 0.28 g($H_2O$)/g(soil). We posit that the higher PAW in kaolinite compared with illite soil arises because of differences in the cumulative pore size distribution.

We measured the pH of all soils containing an equal ratio of water after allowing 1 h to come to equilibrium (Table S2). All quartz-based soils (Q1, Q2a, Q3a) were a neutral pH. In contrast, the pH of the illite (I2a) and montmorillonite (M2a) soils were basic. Because of this alkalinity, I2a and M2a studies were conducted using a growth medium (MIDV1) containing 0.25 M MOPS buffer (pH = 7.0), to separate the effect of mineral addition (e.g., surface area) from pH. Artificial soils supplemented with 1% $CaCO_3$ (Q3a-pH 8) had a pH of 8.51.

Soil surface area ($m^2/g$) was measured using $N_2$ adsorption (Fig. 1E). As quartz particle size decreased, we observed a small increase in surface area (Fig. S2B). In addition, soils containing kaolinite (K2a) and Illite (I2a) presented 10-fold greater surface areas than the most closely related quartz soil (Q2a), while the soil containing montmorillonite (M2a) was 2 orders of magnitude greater than Q2a.

**Effect of particle size on microbial survival.** We hypothesized that microbial growth would vary across soils having different particle size distributions when held at constant water levels since soil matric potential would vary. Specifically, we posited that microbes would grow slower as it becomes harder to pull water from the matrix. To test this idea, we evaluated how particle size affects the growth of *Escherichia coli*, a Gram-negative microbe that is easy to engineer (60), and *Bacillus subtilis*, a sporulating Gram-positive microbe that lives in the rhizosphere (61, 62). To monitor microbial growth, we used microbial strains engineered to produce a rare indicator gas ($CH_3X$). These strains are programmed to constitutively express a plant methyl halide transferase (MHT) that catalyzes the reaction of S-adenosyl-methionine and halide ions to produce volatile $CH_3X$ and S-adenosyl-cysteine (Fig. 2A and B). MHT reporters have been previously used to nondisruptively monitor microbial sensing and gene transfer in soils (27, 28). With this approach, changes in microbial growth and metabolism are monitored by measuring $CH_3X$ in the soil headspace using gas chromatography mass spectrometry (GC-MS).

Growing microbes in our artificial soil matrices required the addition of growth media, but even minimal media formulations contain high nutrient levels that drive the soil osmotic pressure far outside the natural range. For example, M63 minimal medium has an osmotic pressure ($\sim$−1443 kPa) far greater than the normal levels ($\sim$−100–200 kPa) in saturated soils (63). To address this, we diluted M63 medium 16-fold and limited the amount of supplemented NaBr to 20 mM, the minimum required for a strong indicator gas signal (27). This growth medium, designated MIDV1, has an osmotic pressure ($\sim$−319 kPa) close to the normal range found in natural unsaturated soils (64). We characterized the growth of MHT-expressing *E. coli* (*Ec-MHT*) and *B. subtilis* (*Bs-MHT*) in MIDV1 by measuring optical density ($OD_{600}$). *Ec-MHT* grew to a slightly higher $OD_{600}$ than *Bs-MHT* (Fig. S3A and B), albeit at a lower density than observed in the M63 medium. We also evaluated how the indicator gas signal from *Ec-MHT* and *Bs-MHT* relates to CFU. A linear correlation was observed between the $CH_3Br$ signal in the culture headspace and CFU in MIDV1 media (Fig. S4A and B). We also determined $CH_3Br$ partitioning into gas, liquid, and solid phase by measuring standard curves using a chemical standard added to the different artificial soils (Fig. S5).

To evaluate how particle size distribution affects gas reporter activity, we added our gas-reporting microbes ($10^6$ cells) to three soils (Q1, Q2a, Q3a; 800 mg each) held at a constant water content ($\theta$) of 0.25 g($H_2O$)/g(soil). At this $\theta$, the soil matric potential ($\psi_m$) varies across soils. Q1 is above FC ($\psi_m$>−33 kPa), Q2a has a $\psi_m$ = −151 kPa, and Q3a has a $\psi_m$ = −365 kPa. In the sandy artificial soil, *Ec-MHT* and *Bs-MHT* reached a maximum cell density at 20 and 8 h, respectively (Fig. 2C). This duration was slightly faster than that observed in liquid culture (Fig. 2D). In silt loam soils, *Ec-MHT. coli* and *Bs-MHT* reached maximum cell density after a similar period of time as the sandy soil (Fig. 2E) However, in clay soils, *Ec-MHT* required 32 h to reach maximum density, and *Bs-MHT* showed little gas production, suggesting cell death or sporulation (Fig. 2F).

We expected indicator gas production to decrease with time as cell growth slows down due to nutrient consumption. To test this idea, we monitored the rate of $CH_3Br$

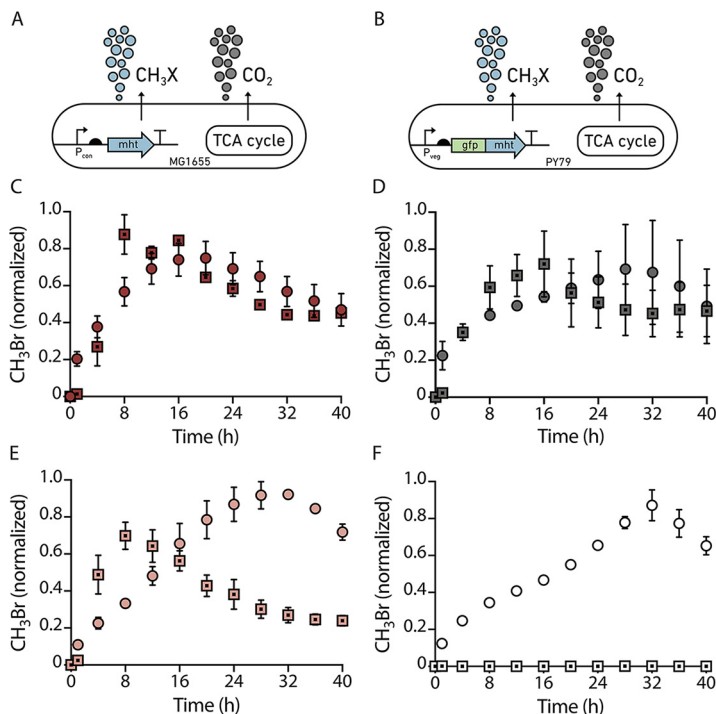

**FIG 2** Effect of particle size distribution on microbial growth. Genetic circuit used to program constitutive indicator gas production in (A) *Ec-MHT* and (B) *Bs-MHT*. In both strains, the MHT gene is chromosomally integrated and expressed using a constitutive promoter so that it is always on. $CH_3Br$ production over time in (C) sand, Q1; (D) liquid; (E) silt loam, Q2a; and (F) clay, Q3a. For each measurement, $10^6$ CFU of *Ec-MHT* (circles) or *Bs-MHT* (squares) in 200 $\mu$L of MIDV1 medium were added to 2 mL glass vials containing 800 mg of soil. Vials were capped and incubated at 30℃. $CH_3Br$ was measured using a GC-MS every 4 h for 40 h. Gas production was normalized to the maximum signal obtained. Error bars represent one standard deviation from three experiments.

production in soils having different particle sizes (Fig. S6A). As expected, gas production from *Ec-MHT* grown in liquid medium decayed exponentially ($R^2$ = 0.71); however, the gas production rate was slower and constant for a longer period of time in silt loam (Q2a) and clay (Q3a) soils. This observation suggests that in soils, matrix particle size plays an additional role in controlling nutrient accessibility and diffusion. Similar experiments were performed with *Bs-MHT* (Fig. S6B). In all conditions, the indicator gas production rate rapidly declined within the first 12 h, suggesting that nutrient deficiency caused cell death or sporulation in this Gram-positive microbe.

**OM-driven changes in soil properties affect microbial growth.** OM can change the physical properties of soil, such as water retention and aggregate stability, but under some conditions, OM also is a food source for soil microbes, making it challenging to separate the physical effects of OM on soil microbial growth. To address this, we grew *E. coli* in soils amended with xanthan and chitin, forms of OM that cannot be metabolized by this microbe. We first measured how the water retention of soils changes with OM addition. Both xanthan and chitin increased the quantity of water retained across the wide range of water potential we measured. The enhancement of water retention increased with OM concentration. Artificial soils with xanthan retained more water (0.5% $_{PAW}$ = 0.42 g[$H_2O$] g[soil]$^{-1}$, 1% $_{PAW}$ = 0.77 g[$H_2O$] g[soil]$^{-1}$) than with chitin (0.5% $_{PAW}$ = 0.50 g[$H_2O$] g[soil]$^{-1}$, 1% $_{PAW}$ = 0.59 g[$H_2O$] g[soil]$^{-1}$) (Fig. S2C). To evaluate if soil water retention properties altered by OM influence microbial growth, we fixed the water content ($\theta$) at 0.4 g($H_2O$)/g(soil). We added *Ec-MHT* ($10^6$ cells) to quartz-based silt loam soils (Q2a) containing 0.5% or 1% (wt/wt) of either xanthan or chitin and measured indicator gas production as a function of time. As controls, we performed experiments in Q2a lacking OM and in liquid medium. In all experiments,

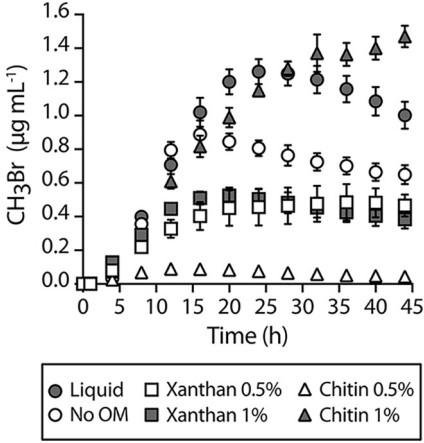

**FIG 3** Effect of OM on microbial growth in soils. *Ec-MHT* ($10^6$ CFU in 200 $\mu$L of MIDV1) were added to 2 mL glass vials containing 800 mg of soils with differentOM source and amount. Vials were capped and incubated at 30°C. $CH_3Br$ was measured using a GC-MS after 0 and 1 h then every 3 h thereafter. $CH_3Br$ production over time in soils with Xanthan (square) or chitin (triangle) at 0.5% (white) or 1% (gray) (wt/wt). An artificial soil without addition of OM, silt loam Q2a soil (white circle), and a liquid control (gray circle) are shown. Experiments were performed in triplicate. Error bars indicate one standard deviation.

we measured $CH_3Br$ accumulation as an indicator of cell growth. Because gas partitioning can be affected by the soil matrix, we generated $CH_3Br$ standard curves in soils containing OM and used them to normalize the data (Fig. S2D). We found that a silt loam soil containing only quartz particles (Q2a) decreased indicator gas accumulation at each time point compared with cells grown in liquid culture (Fig. 3). Addition of xanthan (0.5 or 1% wt/wt) suppressed indicator gas production to a small extent compared with Q2a alone. In contrast, addition of chitin had dramatic effects on indicator gas production. The low level of chitin tested (0.5%) largely suppressed indicator gas production, while the high level led to highest gas production. To better understand growth dynamics, we also calculated the effect of OM on the rate of gas production across the same soils (Fig. S6C). In the soils lacking OM and containing xanthan, peak production occurred after 10 to 15 h, with little production after 20 h. In the soils containing high chitin levels (1%), the peak production occurred at a similar time point. However, gas production continued beyond 20 h, albeit at decreased levels.

**Soil particle size distribution affects signal bioavailability.** In soil, cell-cell communication is mediated by diverse signaling molecules with varied chemistries (65, 66). Signal chemistry and soil properties are thought to modulate signal bioavailability over space and time (67). Biosensors are ideal for studying how signal bioavailability varies in soils as biosensors report on microbial perception at the micron scale (68, 69). To demonstrate the utility of synthetic soils for establishing the mechanisms responsible for modulating signal bioavailability, we studied the bioavailability of acyl homoserine lactones (AHL). This signal is used by bacteria to coordinate population-level behaviors that underlie critical environmental processes, such as greenhouse gas production (70), symbiosis formation (71), and virulence activation (72). These processes are triggered only when the AHL reaches a threshold concentration (73).

To monitor AHL bioavailability in synthetic soils, we used a biosensor that couples gas production to the detection of an AHL with a long acyl chain, 3-oxo-$C_{12}$-HSL (28). In this biosensor (*Ec-AHL-MHT*), transcription of the MHT gene is controlled by the $P_{las}$ promoter. $P_{las}$ is only on when the transcription factor binds to the AHL (Fig. 4A). To obtain a per cell value for indicator gas production, we normalized all $CH_3Br$ measurements to $CO_2$, a gas that correlated to cell number (Fig. S4C and D). We hypothesized that differences in soil water potential would impact AHL diffusion through the soil, thereby limiting microbial access to the signal. To test this idea, we held $\theta$ constant while varying soil particle size, allowing $\psi_m$ to vary as a result of changes in particle

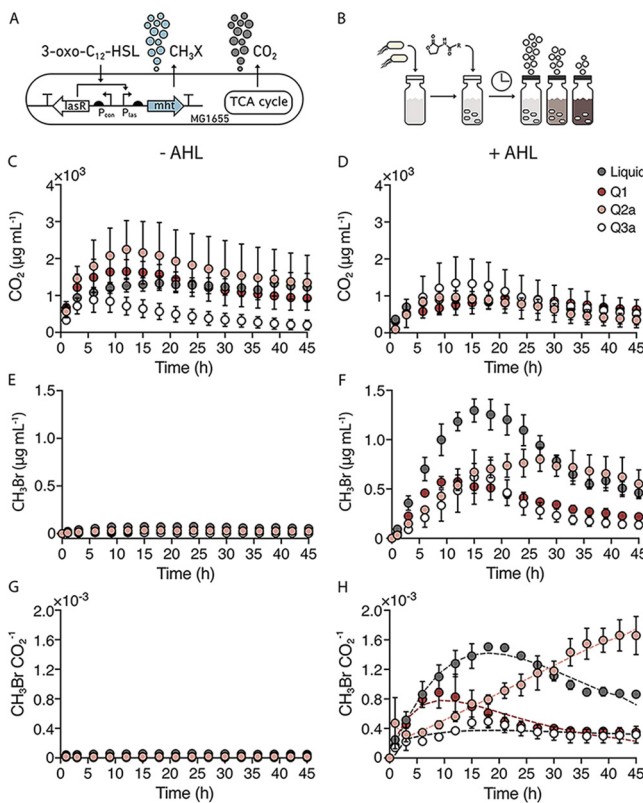

**FIG 4** AHL bioavailability changes with soil particle size. (A) Ratiometric gas reporting approach to monitor cell growth ($CO_2$) and AHL sensing ($CH_3Br$). In this circuit, LasR activates MHT production and $CH_3Br$ synthesis upon binding AHL. (B) To monitor AHL bioavailability, *Ec-MHT* ($10^8$ cells) in MIDV1 medium (100 $\mu$L) were added to the bottom of 2 mL glass vials containing each soil (800 mg). AHL (1 $\mu$M) in MIDV1 medium (100 $\mu$L) was added to the top of the soil, and vials were capped and incubated at 30°C. $CH_3Br$ and $CO_2$ were measured using GC-MS at time zero and 1 h after capping, and then every 3 h. (C) $CO_2$ production in the absence and (D) presence of AHL reveals that cells grow under both conditions. (E) $CH_3Br$ production in the absence and (F) presence of AHL reveals that indicator gas production is AHL-dependent. (G) The ratio of $CH_3Br/CO_2$ allows for a comparison of the AHL sensed per cell in the presence and (H) absence of AHL. This data shows that soil texture affects the dynamics of AHL sensing. The dashed lines represent a fit to an exponential growth-decay model. Dots indicate average and error bars indicate one standard deviation calculated with $n = 3$.

size. To accomplish this, we varied soil particle sizes (using soils Q1, Q2a, and Q3a) held at a fixed $\theta$ of 0.25 g($H_2O$)/g(soil). We added *Ec-AHL-MHT* ($10^8$ cells) at the bottom of 2 mL vials containing the artificial soils or no matrix (liquid control). After a 30-minute incubation, we added water (100 $\mu$L) containing or lacking AHL (1 $\mu$M) to the top of the soil (Fig. 4B). We then capped the vials and monitored $CH_3Br$ (dependent on microbial AHL detection) and $CO_2$ (the proxy for cell growth) accumulation at different time points. We observed biosensor respiration in all soils ±AHL (Fig. 4C and D). In the absence of AHL, the indicator gas signal was low (Fig. 4E) as well as the ratiometric signal (Fig. 4G). In contrast, addition of AHL (1 $\mu$M) led to a time-dependent indicator gas signal in all soils (Fig. 4F). Normalization of this signal to respiration allowed for direct comparison of sensing in different soils (Fig. 4H). To quantify differences in AHL sensing dynamics, the ratiometric ($CH_3Br/CO_2$) data was fit to an exponential growth and decay model to calculate the maximum $CH_3Br/CO_2$ signal ($A$) and the time for half maximum gas accumulation ($T_{1/2}$) (74). We observed that $A$ and $T_{1/2}$ both vary with soil particle size. Maximum gas accumulation ($A$) follows the trend: liquid $\approx$ silt loam (Q2a) > sand (Q1) > clay (Q3a). In sand (Q1), the AHL bioavailability was 55% $\pm$ 12% that observed in liquid, while in silt loam (Q2a) and clay soils (Q3a), it was 94% $\pm$ 9% and 26% $\pm$ 2%, respectively. The speed of signal transmission also varied between artificial

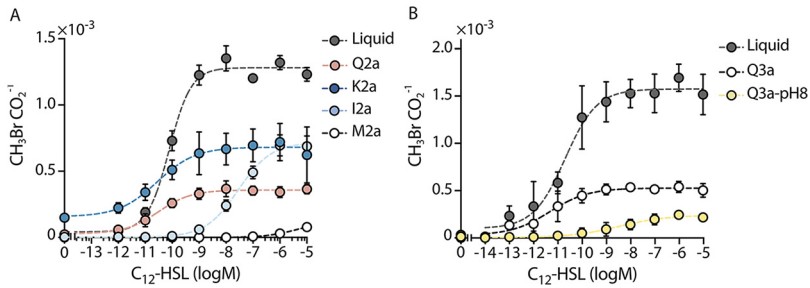

**FIG 5** Mineralogy and pH affect AHL bioavailability. (A) AHL bioavailability in soils with different mineralogy but the same texture. *Ec-AHL-MHT* ($10^8$ CFU) were mixed with different concentrations of AHL and immediately added to vials containing artificial soils with different clay types in MIDV1. The medium contained 0.25M MOPS to obtain a $\psi_m = -80$ kPa. Vials were capped, and $CH_3Br$ and $CO_2$ were measured using a GC-MS after 6 h. The dashed lines represent a Hill function fit to the data. With this fit, different $k$ values were obtained for liquid ($7.8 \times 10^{-11}$), Q2a ($3.1 \times 10^{-11}$), K2a ($2.8 \times 10^{-11}$), I2a ($2.9 \times 10^{-8}$), and M2a ($\geq 4.2 \times 10^{-6}$). (B) Different amounts of AHL were added in 100 $\mu$L of MIDV1 medium to vials containing artificial soils with different pH. AHL was incubated for 30 min in the soils before adding the AHL biosensor ($10^8$ cells) in 100 $\mu$L of MIDV1 containing 0.25 M MOPS, pH 7.0 to achieve FC. Vials were capped and gas production was measured after 6 h. The $CH_3Br/CO_2$ ratio represents the per cell sensing of AHL. The dashed lines represent a Hill function fit to the data. With this fit, distinct $k$ values are obtained with liquid ($2.0 \times 10^{-11}$), Q3a ($4.7 \times 10^{-12}$), and Q3a-pH 8 ($3.5 \times 10^{-9}$). Error bars represent one standard deviation determined from three experiments.

soils, with sand (Q1; $T_{1/2} = 1.8 \pm 0.7$ h) allowing faster transmission than silt loam (Q2a; $13.8 \pm 2.3$ h) and clay (Q3a; $3.7 \pm 0.6$ h).

To better understand the dynamics of AHL sensing, we also calculated the $CH_3Br/CO_2$ production rate to visualize how the gas accumulation trends relate to the duration the microbes were growing (Fig. S7). As expected, the ratiometric $CH_3Br/CO_2$ signal in liquid follows an exponential decay, which we posit is driven by nutrient consumption ($R^2 = 0.73$). The rate of gas production was highest in sand (Q1) and silt loam (Q2a) and lowest in clay (Q3a). However, gas production persisted longer in silt loam (Q2a) and clay (Q3a).

**Soil mineralogy and pH affect signal bioavailability.** Prior studies have shown that AHLs can sorb onto some soils (28, 75), but it is not known how soil mineral composition affects signal sorption. We hypothesized that differences in mineral surface area will impact the amount of AHL sorbed into the matrices, thereby changing microbial perception of this signal. To test this idea, we evaluated AHL bioavailability within soils having the same particle size distribution (silt loam) but different mineralogy (different surface area), including quartz only (Q2a), quartz and kaolinite (K2a), quartz and illite (I2a), and quartz and montmorillonite (M2a). Because water retention and pH vary with mineralogy, and pH can affect AHL half-life (29), we kept the $\psi_m$ fixed at $-80$ kPa and used buffered MIDV1 medium containing 0.25 M MOPS (pH = 7). For these experiments, we incubated *Ec-AHL-MHT* ($10^8$ cells) with varying AHL concentrations in soils and liquid for 6 h and then measured $CH_3Br$ and $CO_2$ (Fig. 5A). We chose this incubation time because *E. coli* showed similar growth across the different soils (Fig. S8). We found that soil containing 2:1 clay increased the AHL concentration needed to achieve a half maximum indicator gas production ($k$) by up to 3 orders of magnitude for a non-expanding clay (I2a) and shifted the response curve outside of the range of testing, indicating that the $k$ value was changed by at least 5 orders of magnitude for an expanding clay (M2a) compared with soils made of quartz (Q2a). Surprisingly, addition of a 1:1 clay (K2a) did not affect $k$, even though the K2a surface area is ~10-fold higher than Q2a. We interpret this last result as arising because of differences in the porosity and/or cation exchange capacity of the 1:1 clay (K2a) compared with quartz (Q2a).

Prior studies in liquid culture have shown that the AHL lactone ring can undergo a pH-dependent hydrolysis reaction (76). In the absence of a lactone ring, microbes can no longer use this signal to communicate with one another (29). To explore whether soils with different pH values affect AHL bioavailability *in situ*, we performed experiments in two clay-sized, quartz-based soils. One soil had a neutral pH (Q3a), while the

**TABLE 1** Physicochemical properties of the Mollisol[a]

| Natural soil | N (%) | TC (%) | OC (%) | Sand (%) | Silt (%) | Clay (%) | SA (m$^2$/g) | pH |
|---|---|---|---|---|---|---|---|---|
| **Avg** | 0.1 | 8.5 | 1.1 | 13.6 | 31.3 | 55 | 30.9 | 8.7 |
| **Error** | 0 | 0.1 | 0.1 | 0.5 | 1 | 0.5 | 0.8 | 0.08 |

[a]Total nitrogen (N%), total soil carbon (TC%), organic carbon content (OC%), percentage of particle size (sand, silt, clay), surface area (SA), and pH of the natural soil from the USDA Grassland Soil and Water Research Lab in Temple, TX.

other (Q3a-pH 8) was adjusted to pH = 8.5 using $CaCO_3$ (Table S2). For these experiments, we first incubated different concentrations of AHL in the soils for 20 min. We then added *Ec-AHL-MHT* (10$^8$ cells) in buffered media to field capacity ($\psi_m = -33$kPa) to homogeneously mix the buffer in the soil; this led to identical pH values across samples. After 6 h, we measured indicator gas production (Fig. 5B). We found that microbes required ~3 orders of magnitude more AHL in alkaline soils (2.25 nM) than neutral soils (3.42 pM) to trigger a similar population-level indicator gas production.

**Comparing natural and artificial soils.** As a first test of how individual soil properties affect soil microbial behavior, we generated a new series of artificial soils that sequentially recreated the properties of a natural soil, and then monitored one microbial behavior (AHL detection) as artificial soil complexity increased. Our artificial soils were designed to replicate properties of the A horizon of an Austin Series Mollisol (Udorthentic Haplustoll), which we collected from the USDA facility in Temple, Texas. This Mollisol is an alkaline silt clay loam containing a 2:1 clay type (see Table 1 for N content, organic carbon content, pH, and particle size distribution). Each artificial soil in our series differed by one layer of complexity from the natural Mollisol (artificial soil physical properties reported in Fig. S2E). To understand how soil properties interact to modulate signal bioavailability, we incubated the AHL biosensing microbes in our artificial soil series with different amounts of AHL at a constant $\theta = 0.4$ for 6 h. We then measured $CH_3Br$ and $CO_2$, generated AHL dose-response curves, and fitted each curve to a Hill function (Fig. 6A). We found that as artificial soil complexity increased, natural and artificial soil AHL biosensor response curves grew more closely matched.

In examining the effects of both natural and artificial soil matrices on microbial behavior, we explored two parameters describing microbial response: maximum biosensor gas production (A), expressed as $CH_3Br/CO_2$ (Fig. 6B), and the amount of AHL required to generate half maximum signal induction (k), expressed in pM (Fig. 6C), both terms generated from Hill equation fitting. We used A to understand matrix effects on the magnitude of biosensor signal output per cell, and we used k to understand the effects of matrix sorption on the shape of biosensor signal response. First, comparing A from biosensors grown in the natural Mollisol to A generated by biosensors grown in liquid media, we found the response to AHL was 10.7 times lower in the Mollisol (Fig. 6B). The synthetic soil series gives us insight into the parameters driving this change: comparing the synthetic soil series A values to those generated by biosensor growth in liquid media, we found the largest change in AHL response was driven simply by the addition of a physical matrix (quartz particles), mimicking only the particle size distribution of the natural soil ($P < 0.001$). In the quartz-only artificial soil the maximum gas production (A) decreased by half compared to liquid media. These results underline the importance of the physical matrix in controlling microbial accessibility of diffusible compounds. Adding mineralogy and pH (PS+M+pH) to match the Mollisol content also significantly changed the maximal gas production from particle size (PS) alone ($P < 0.03$), although particle size and mineralogy (PS+M) were not significantly different from PS alone. Finally, soil containing organic matter (PS+M+pH+OM) was not significantly different from PS+M+pH, PS+M, or PS. However, it was significantly different from the Mollisol ($P < 0.03$).

To determine if the AHL concentration needed for half maximum induction was influenced by soil composition, we evaluated k (Fig. 6C). We found that the liquid control and the artificial soil that only resembles the natural soil's particle size (PS) have k

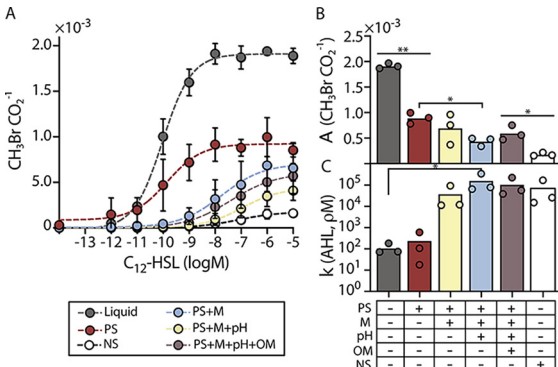

**FIG 6** AHL bioavailability in artificial soils that recreate different properties of a Mollisol. (A) Different concentrations of AHL were added to MIDV1 medium (300 $\mu$L) containing *Ec-AHL-MHT* ($10^8$ CFU), and this mixture was mixed with a series of artificial soils (700 mg) that mimic different levels of complexity found in a Mollisol from Austin, TX. $CH_3Br$ gas was normalized by the $CO_2$ signal measured after a 6-h incubation in closed vials. The dashed line indicates the Hill function fit to the data. PS = particle size, M = mineralogy, pH = addition of $CaCO_3$, OM = addition of xanthan gum, and NS = natural soil. In the case of PS+M soil, 0.25 M MOPS, pH 7.0 was included in the buffer to isolate the effect of mineralogy on bioavailability. (B) Maximum gas production ($CH_3Br/CO_2$) obtained from a fit of the data to the Hill equation reveals a decrease in maximum gas production as soil complexity increases (ordinary one-way ANOVA, Tukey's multiple comparisons; **, $P < 0.001$, *, $P < 0.03$). (C) Amount of AHL (pM) necessary for a half maximum gas response. Artificial soil that recreates texture, mineralogy, and pH requires an AHL concentration within the same order of magnitude as the natural soil to induce the biosensor (nonparametric ANOVA, Dunn's multiple comparisons; *, $P < 0.03$). Error bar represents one standard deviation calculated from three replicates.

values that differ by <2.5-fold (~110 pM and ~244 Pm, respectively), while adding mineralogical complexity (PS+M) increases $k$ by at least 2 orders of magnitude (~39 nM). This result suggests that sorption of the AHL onto the mineral surfaces in the Mollisol decreases AHL bioavailability. Soil pH (PS+M+pH) further increased $k$ by 1 order of magnitude (~163 nM) suggesting that pH-dependent hydrolysis also decreases AHL bioavailability in the Mollisol. The $k$ values observed in liquid culture and PS+M+pH experiments were significantly different ($P < 0.03$). The OM source (xanthan gum) used to make artificial soils (T+M+pH+OM) did not influence AHL bioavailability. Finally, the AHL concentration required for half maximum induction in the Mollisol (~107 nM AHL) was within 2-fold of the value observed in the artificial soil built to mimic the natural soil's particle size, mineralogy, and pH.

## DISCUSSION

We designed a flexible recipe for artificial soils that allows for the individual, linear variation of a number of soil-forming factors. These artificial soils are an experimental tool built to allow researchers to study the impacts of each individual soil property on a whole soil's behavior. They should be potentially useful in a wide range of experiments probing the chemical, physical, and biological effects of individual soil properties on the entire soil system. We demonstrated that microbial growth and signaling can be monitored *in situ* using these constructed soils. We additionally tested the generality of one underlying assumption of the artificial soil design: that some soil properties present additive behaviors when starting with texture as the simplest component evaluated. To track microbial response, we used biosensors, which are microbes that report on their environment and/or their behaviors.

Although biosensors have been used to measure bacterial growth in soils (27, 28), previous studies used incubation conditions to support optimal biosensor performance, such as excess of nutrients and salts. In this work, we show how biosensors can be used under experimental conditions that mimic natural soil osmotic conditions, setting water potential in the artificial matrices to values within real soil water potential ranges. By building artificial soils that vary two properties (particle size and OM content)

separately, we benchmarked the effects of these variables individually on microbial growth. We observed a correlation between growth and soil water potential in artificial soils sampling a range of particle sizes held at similar water contents (Fig. 2). *E. coli* showed consistent maximal growth in soils with a range of particle sizes (Q1, Q2a, and Q3a), with peak growth occurring at later time points as particle size decreased and water potential increased. This finding suggests diffusional control on growth. *B. subtilis* showed peak growth at a similar time point in sand (Q1) and silt loam (Q2a) soils but did not grow in the clay soil (Q3a). The lack of *B. subtilis* growth at the highest water potential values suggests that sporulation was triggered under these conditions.

The addition of OM to silt loam (Q2a) caused cell growth to vary with the source of OM. Overall, the complexity in cell growth response to OM suggests nonlinearities in the effects of OM on cell growth. Neither source of carbon used here (xanthan or chitin) was metabolizable by the microbes used; therefore, these experiments tested only the changes in microbial growth driven by OM physicochemical effects on soils. Increasing xanthan levels triggered a large increase in soil osmotic potential (Fig. 3), which decreased cell growth to a small extent, regardless of the amount added. In contrast, 1% and 0.5% chitin did not significantly alter osmotic potential but suppressed cell growth at 0.5% chitin and enhanced growth at 0.1% chitin. The OM effects on microbial growth do not follow a simple model where changes in matric potential linearly alter cell growth. These growth trends suggest that changes in osmotic potential do not alone control growth in these soils and that OM controls on cell growth are complex, even in studies that use simplified materials that cannot be metabolized.

It is possible to conceive future experiments to deconvolve the nonlinearities observed here. In future cell growth experiments, other sources of OM (e.g., plant-derived OM, such as mucilage) can be added to the artificial soils (77). Biosensors could also be incubated in the presence of microbial communities capable of metabolizing different OM sources to evaluate differences in water and nutrient availability simultaneously (78). A complementary approach would be to incorporate microbial communities for longer incubations, allowing them to grow and form biofilms, and naturally microaggregate the synthetic matrix, thereby dynamically changing the soil OM content. Biosensors could be incorporated at different time points to report changes in water retention properties. A potential limitation for the latter approach is the large number of biosensor cells ($10^6$–$10^8$) required for a detectable gas signal, which could artificially shift the abundance of microbial community members. To increase the sensitivity of gas detection while using fewer cells for incubation, gas preconcentration could be performed prior to GC-MS analysis (79). While this approach has not been applied to gas biosensors, it is widely used to measure emissions of methyl halides from fields (80).

Our biosensor studies within artificial soil show that successful delivery of the microbial AHL signal is controlled by a combination of soil texture, mineralogy, and pH. Prior studies examining quorum sensing in microfluidic devices suggest that fluid flow affects AHL bioavailability (81). Since water and oxygen content are controlled by soil texture when water content is held constant, we compared AHL signal transmission through quartz-based matrices that sample a range of particle sizes. We hypothesized that the major determinants of signal dynamics under these experimental conditions are nutrient and signal flow through the matrix, altering microbial growth and signal availability. We found that the maximum AHL bioavailability varied by 3.6-fold, and the speed of signal transmission varied by 7.7-fold (Fig. 4), with sandy soils leading to more rapid signal transmission and silty soils leading to greater overall signal transmission. To further expand on these initial findings, biosensor experiments need to be performed at larger scales, for example, using soil columns coupled with probes to measure water and gas flow.

Mineralogy and pH had more dramatic effects on signal bioavailability than particle size, with mineralogy triggering AHL bioavailability shifts of $10^5$ and pH causing shifts of $10^3$, compared to particle size shifts in AHL bioavailability, which were less than

10-fold (Fig. 5) In other studies, biochar surface area has been found to correlate with loss of bioavailable AHL (30). Similarly, we found that artificial soils that vary surface area by changing mineralogy also decrease AHL bioavailability when held at the same pH. Our artificial soil measurements showed that mineralogy could decrease the AHL concentration by at least 5 orders of magnitude, implicating sorption as a significant mechanism for attenuating AHL signals. The magnitude of this effect exceeds that previously observed when biochars having a range of surface areas were mixed with AHLs in a petri dish (30). Furthermore, alkaline artificial soil decreased AHL bioavailability by 3 orders of magnitude. This effect is consistent with petri dish studies showing that the half-life of the lactone ring decreases in alkaline solutions because of hydrolysis (29). Our results suggest that AHL half-life, which influences calling distance, will vary by many orders of magnitude across different soils. In future studies, it will be interesting to sample soils with an even more comprehensive range of pH and mineralogy properties and determine if these physicochemical properties are also essential factors impacting other microbial signals' persistence, such as short chain AHLs and peptide autoinducers.

Our measurements comparing AHL bioavailability in a Mollisol and within artificial soils that mimic different levels of complexity within the Mollisol revealed that the primary factor affecting maximum gas production is particle size distribution. In addition, the AHL concentration needed for half-maximum induction of the biosensor output was attenuated to the greatest by the addition of soil mineralogy and pH. These results illustrate how artificial soils will be useful in future studies to give mechanistic information on how different matrix properties affect biological activity. As such, artificial soils will be useful for informing predictive models to better understand natural soils' influence on biological processes. Herein, we found that the addition of microbially-derived OM to artificial soils does not bring the artificial system closer to a natural system. This result does not invalidate the possibility that other sources of OM might play a role attenuating AHL. An explanation for OM's nonlinear behavior is consistent with prior studies implicating soil organo-mineral associations as an emergent property (82). The formation of natural organo-mineral associations is a product of long-term soil formation processes, including the decade to millenial-scale interactions of plant, animal, and microbial exudates with soil minerals. Mimicking these processes within our simplified artificial soil model will require more methods development.

This study used artificial soils in small gram-scale incubations, but they can be compatible with larger testing platforms, such as 3D-printed devices like EcoFAB (83, 84), soil boxes (56), and other climate-controlled chambers (85). These artificial soils can be combined with other tools developed to simplify soil microbiology studies to understand microbial processes, such as standardized soil growth media (86) and simplified soil microbial communities (87, 88). Stable microbial communities with reduced complexity are particularly appealing to target for future studies (89), since these soil consortia allow for studies of interspecies interactions that control community-scale behaviors, which can be intractable in native soil microbiomes.

This study also shows that synthetic microbes that function as biosensors are compatible with synthetic soils. The use of synthetic soils and biosensors is expected to be useful for testing hypotheses generated by high throughput omics techniques implemented in systems biology (69). This work focused on evaluating cell growth and perception of AHL signals, but future studies can use artificial soils to study other dynamic biological processes that are controlled by the bioavailability of different chemicals (Fig. 7A). The biosensors developed here can be easily used to report how metabolically active a microbe is in various soil and hydration conditions (Fig. 7B). Using constitutive biosensors in different artificial soils makes it possible to study microbial survival and distribution in a mixed community and shed light on biogeochemistry questions such as the effect of hydration pulses on soil respiration (90, 91) (Fig. 7B). Additionally, gas biosensors could be coded to report on the bioavailability of other environmental parameters, such as different signaling molecules, intermediates in biogeochemical cycles, metal

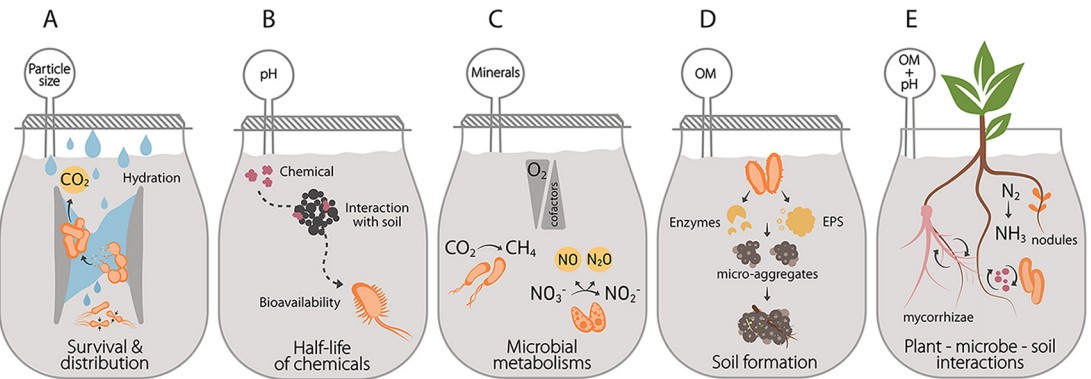

**FIG 7** Artificial soil applications. Artificial soils with different properties can be used to study (A) the bioavailability of a wide range of chemicals of interest, (B) the survival and distribution of microorganisms under different hydration conditions, (C) microbial metabolisms under different oxygen gradients and availability of cofactors, (D) soil formation, and (E) plant-fungi-bacteria interactions.

ions critical to the biocatalysis underlying these cycles, and osmolytes critical to survival in harsh environmental conditions (69). By designing artificial soils to deconvolute soil heterogeneity, in other words, artificial soil series where one property gradually changes, it will be possible to explore the effect of soil redox and nutrient gradients on microbial metabolism that regulates the outcome processes relevant at planetary scales (Fig. 7C), such as the production of greenhouse gases (92, 93). As explored in other studies (21, 94), artificial matrices can be used in long-term experiments to study how different variables (e.g., microbial communities and mineral-organic interactions) can affect soil formation (Fig. 7D). Finally, artificial soils are a helpful platform to study plant, fungi, and bacterial dynamics such as root colonization (95), agonist and antagonist interactions (96), and how different types of soil amendments and land management practices impact these interactions (Fig. 7e).

## MATERIALS AND METHODS

**Soil materials.** Whole grain fine quartz sand (NJ2, particle size $\sim$70 $\mu$m), and ground silt-sized quartz (Min-U-Sil40, particle size $\sim$8.71 $\mu$m), and clay-sized quartz (Min-U-Sil5, particle size $\sim$1.7 $\mu$m) were from US Silica. Kaolinite clay (one-layer octahedral sheet and one-layer tetrahedral sheet) and montmorillonite clay (two layers of tetrahedral sheets and one layer of octahedral sheet) were from Spectrum Chemical MFG Corp. $CaCO_3$ powder was from Acros Organics, glass vials (2 mL) and caps used for soil incubations were from Phenomenex, and 3-oxo-$C_{12}$-HSL was from Sigma-Aldrich. Other chemicals were from Thermo Fisher Scientific, Millipore, or Sigma-Aldrich.

**Artificial soil production.** A total of seven artificial soils were created using the protocol provided in the supplemental materials. These matrices, which sampled three common soil textures [sand (Q1), silt loam (Q2), and clay (Q3)] were created by mixing sand, silt, and clay-sized quartz as shown in Table 2. To do this, quartz materials (150 g total) were added to an autoclaved 250 mL wide-mouth volatile organics analysis (VOA) glass jar (Thermo Fisher V220-0125). Samples were first manually mixed for 30 s by shaking and repeatedly turning the jar, and then placed on a horizontal shaker (VWR OS-500) at 7 rpm for 30 min. To design artificial soils that simulate reactive clay minerals, the clay-sized quartz fraction in the aggregated quartz-based silt loam artificial soil (Q2a) was replaced with clay minerals (Table 2). Kaolinite (K2a), illite (I2), and montmorillonite (M2a) were used because they represent the major structural classes of soil minerals. To create a higher soil pH, $CaCO_3$ power (0.5% wt/wt) was added to dry soil of the same texture and mineral composition as Q3a to create Q3a-pH 8. Additionally, we designed four soils to contain model organic compounds to mimic soil organic matter. To this purpose, we thoroughly mixed xanthan or chitin at 0.5% and 1% wt/wt with dry artificial soils mixtures.

Soil aggregate structure was created by subjecting the soils to multiple wet-dry cycles. In a glass jar, megaOhm water was mixed to reach water holding capacity (WHC). The mixture was stirred using a spatula until all grains were hydrated and a slurry-paste formed. The paste was poured into an aluminum pan and dried in an oven at 60°C overnight. Dry material was gently broken using a spatula into fragments small enough to be placed in the original glass jar. This procedure was repeated twice. The synthetic, aggregated matrices were sieved using a series of U.S. standard sieves. Aggregates ranging from 0.85 to 1.44 mm were used in all studies. We autoclaved matrices twice before use to sterilize. Q1 was not subject to this protocol because it does not aggregate.

**Water retention curves.** Water retention was characterized at room temperature using a WP4C dew-point potentiometer (Decagon Devices, Inc.). The typical accuracy of the equipment is $\pm$50 kPa, and

**TABLE 2** Composition of the artificial soils

| Soil name | Q1 | Q2a | Q3a | K2a | I2a | M2a |
|---|---|---|---|---|---|---|
| Mineralogy | Quartz | Quartz | Quartz | Kaolinite | Illite | Montmorillonite |
| Texture | Sand | Silt loam | Clay | Silt loam | Silt loam | Silt loam |
| **Composition (wt/wt)** | | | | | | |
| Quartz sand | 90 | 20 | 20 | 20 | 20 | 20 |
| Quartz silt | 5 | 60 | 20 | 60 | 60 | 60 |
| Clay-sized quartz | 5 | 20 | 60 | 0 | 0 | 0 |
| Kaolinite | 0 | 0 | 0 | 20 | 0 | 0 |
| Illite | 0 | 0 | 0 | 0 | 20 | 0 |
| Montmorillonite | 0 | 0 | 0 | 0 | 0 | 20 |

therefore the WP4C is generally not ideal for wet samples with water potential $>-100$ kPa. Briefly, Milli-Q water was added to each artificial soil in a glass jar and thoroughly mixed. The jar was covered and allowed to equilibrate for 2 to 4 h. Artificial soil ($\sim$5 g) was transferred to a stainless steel sample cup and analyzed using the WP4C in continuous mode. Under these conditions, the accuracy of the instrument is improved to $\pm$10 kPa (97). After the measurement, the sample was dried in the oven overnight at 60°C, and the dry weight was recorded. The water content ($\theta$) was calculated based on the water loss following drying. Before each sample measurement, the instrument was calibrated with a 0.5 M KCl standard solution. Water retention data was fit to the van Genuchten model (58) as shown in equation 1:

$$Se = \left[\frac{1}{1+(\alpha h)^n}\right]^m, \left(m = 1 - \frac{1}{n}\right)$$

where $Se$ corresponds to the effective water content defined as,

$$Se = \frac{(\theta - \theta_r)}{(\theta_s - \theta_r)}$$

and a SWRC nonlinear fitting program was used to find $\theta_s$, $\theta_r$, $\alpha$, $n$ from the experimental data as described in Seki, 2007 (59).

**Surface area analysis.** Soil surface area was measured using a Quantachrome Autosorb-3b Surface Analyzer. Samples were degassed in glass cells and vacuum dried overnight at 200°C. Nitrogen adsorption/desorption isotherms were obtained at 77 K by using a 26-point analysis to obtain the relative pressures $P/P_0$ from $1.21 \times 10^{-4}$ to 0.99, where P is the adsorption equilibrium pressure and $P_0$ is the vapor pressure of bulk liquid $N_2$ at the experimental temperature. The specific surface area was calculated using Brunauer-Emmett-Teller (BET) theory.

**Microbial strains.** Three strains were used for these studies. To study soil effects on growth, we used a previously developed *Escherichia coli* MG1655 strain (designated *Ec-MHT* here) that constitutively expresses MHT from a chromosomally incorporated reporter gene (27). A *Bacillus subtilis* PY79 strain was also generated that constitutively expresses a fusion of green fluorescent protein (GFP) and MHT. This strain (designated *Bs-MHT*) was created by building DNA that uses a strong constitutive promoter ($P_{veg}$), a ribozyme (RiboJ), and RBS (MF001) to express a gene encoding a GFP-MHT fusion; the plasmid contains the *ermB* gene, which confers erythromycin resistance. This DNA, which was flanked by sequences that exhibit homology to the *ganA* (a $\beta$-galactosidase) region of *B. subtilis* PY79 chromosome, was cloned into a shuttle vector containing a kanamycin resistance cassette and a pSC101 origin of replication using Golden Gate cloning (98). The sequence-verified construct was linearized and transformed into *B. subtilis* (99). Chromosomal integration was confirmed using PCR on the purified genome. GC-MS was used to show that this strain (PY79-*mht*) constitutively produces $CH_3Br$ like *Ec-MHT*. These strains both represent indicator gas reporters of microbial growth and metabolism, since their synthesis of methyl halides can be used to track their relative growth in a variety of hard-to-image environments. To study the bioavailability of AHLs, we used a previously described *E. coli* MG1655-las biosensor (designated *Ec-AHL-MHT* here) that reports on the presence of 3-oxo-$C_{12}$-HSL by synthesizing $CH_3Br$ (28). With this latter strain, methyl halide production is dependent upon detection of 3-oxo-$C_{12}$-HSL, and normalization of this gas signal to respiration allows for a per cell signal that can be related to the HSL detected by the cell population.

**Growth medium.** Lysogeny Broth (LB) containing 10 g/L tryptone, 5 g/L yeast extract, and 10 g/L sodium chloride was used for culturing strains and all engineering. For soil studies, we developed a modified M63 minimal medium that we called MIDV1, which has an osmotic pressure within the range found in natural soils. MIDV1 contains 1 mM magnesium sulfate, 0.2% glucose, 0.00005% thiamine, 0.05% Casamino Acids, 20 mM sodium bromide, and 0.0125% of the M63 salt stock. The M63 salt stock solution was generated as described in Cheng et al. (28). The M63 salt stock and the water were autoclaved before use, while all other components were sterile filtered. For mineralogy and pH experiments, we added 0.25 M 3-(*N*-morpholino) propanesulfonic acid (MOPS) to the MIDV1 media to buffer the soil pH. MOPS has a p$K_a$ of 7.2 and is commonly used to buffer bacterial growth medium (100).

**Cell survival assay.** To analyze microbial survival in artificial soils, matrices with different particle size distributions (Q1, Q2a, and Q3a) were adjusted to a water content ($\theta$) of 0.25 g(H$_2$O) g(soil)$^{-1}$. Single colonies of *Ec-MHT* or *Bs MHT* were used to inoculate LB cultures (6 mL) and grown for 18 h while shaking at 37°C and 30°C, respectively. Cultures were diluted 100× into LB medium and allowed to grow to mid exponential phase (OD$_{600}$=0.5). Cells were pelleted by centrifugation (3000 rpm, 10 min), and resuspended in MIDV1 two times. Cells (10$^6$ CFU in 200 $\mu$L) were added into glass vials (2 mL) containing artificial soils or lacking a matrix. Vials were capped, incubated at 30°C, and the total CH$_3$Br accumulation was analyzed after different incubation times. To allow comparison between strains, CH$_3$Br ($\mu$g/mL) was normalized to the maximum signal observed across different soil types. The CH$_3$Br production rates were calculated by subtracting gas production between adjacent data points. In the case of the liquid growth experiment, the gas signal was fit to equation 2:

$$y = (y_0 - \beta)e^{-Kt} + \beta$$

where $y_0$ is the initial signal at $t = 0$, $\beta$ is the maximum signal, and $K$ is the rate constant. To analyze microbial survival in artificial soils containing OM, experiments were repeated using a similar protocol with quartz-based silt loam soil (Q2a) lacking or containing xanthan or chitin at 0.5% and 1% (wt/wt). Soils were adjusted to a water content ($\theta$) of 0.4 g(H$_2$O) g(soil)$^{-1}$.

**Signal bioavailability assays.** To study AHL diffusion variability, we added cells to artificial soils (Q1, Q2a, and Q3a) at a low $\theta = 0.25$ g(H$_2$O) g(soil)$^{-1}$. *Ec-AHL-MHT* were cultured as described for *Ec-MHT* in growth assays. *Ec-AHL-MHT* (10$^8$ CFU in 100 $\mu$L) was added to glass vials (2 mL) containing the artificial soils. A vial containing only the nutrients (no matrix) was included as a control. We sealed the vials using a hydrophobic porous sealing film and incubated them for 30 min at 30°C in a static incubator. After the incubation, we added AHL (1 $\mu$M) or sterile water (control) to reach a $\theta = 0.25$ g(H$_2$O) g(soil)$^{-1}$. Each vial takes 3 min to be analyzed using the GC-MS. For this reason, we capped one vial every 3 min to accumulate the gas for the same amount of time, and sequentially measured the total gas accumulation in vials every 3 h using the GC-MS for a total of 45 h.

We fitted the AHL pulse in soils after normalization (see statistics section) to a model using a multiplication of two Kohlraush functions, or stretched exponential functions as described in Peleg et al. (74) and shown in equation 3:

$$N(t) = N_o \exp\left[\left(\frac{t}{t_{cg}}\right)^{m1}\right] \exp\left[\left(-\frac{t}{t_{cd}}\right)^{m2}\right]$$

where $N_o$ is the initial strength of the signal, the variables $t_{cg}$ and $t_{cd}$ represent the characteristic times for the signal growth and decay, respectively. We used the parameters specified in Munoz-Lopez et al. (101) for equation 4:

$$N(t) = N_o \exp(\alpha t^\beta) \exp(-at^b)$$

where $\alpha = \left(\frac{1}{t_{cg}}\right)^{m1}$, $a = \left(\frac{1}{t_{cd}}\right)^{m2}$, $\beta = m_1$, $b = m_2$, are outcomes of the fitted model, we calculated the time at which maximum signal ($T_{max}$) is achieved in each soil by maximizing the model using the minimize Nelder-Mead algorithm. We calculated the half-maximum induction ($T_{\frac{1}{2}max}$) by solving $\frac{T_{max}}{2}$ when the initial guess ($x_0$) $x_0 < T_{max}$. The CH$_3$Br production rate was obtained by subtracting the total gas accumulated between adjacent data points.

We tested the bioavailability of AHL in artificial soils that have the same particle size distribution and aggregation, but different mineralogy (Q2a, K2a, I2a, and M2a). We first fixed the soil water potential ($\psi$) to $-80$ kPa using data from water retention curves (WRC) for each soil. We mixed *Ec-AHL-MHT* (10$^8$ cells) in 200 $\mu$L of MIDV1 containing 0.25 M MOPS, pH 7.0 with sterile water until the water content for the fixed soil water potential (–80 kPa) was achieved. We added 10× dilutions of 3-oxo-C$_{12}$-HSL to the cells in a total volume of 1 $\mu$L of dimethyl sulfoxide (DMSO) and immediately transferred them into 2 mL glass vials containing the different artificial soils. A vial containing only the nutrients (no matrix) was included as a control. We capped the vials every 3 min and accumulated gas at 30°C in a static incubator. We measured the gas production after 6 h using GC-MS. We fitted the gas production data to a dose-response curve after normalization using the Hill function shown in equation 5:

$$y = b + \frac{A \times x^n}{k^n + x^n}$$

where $y$ represents gas production by the biosensor, $x$ is the AHL concentration, $A$ is the maximal response, $b$ is the basal response, $n$ indicates the steepness of the curve, and $k$ is the AHL concentration that produces the half-maximum response.

Experiments measuring AHL bioavailability in soils with different pH (Q3a and Q3a-pH 8) were performed at a fixed $\psi = -33$ kPa. Dilutions of 3-oxo-C$_{12}$-HSL in 1 $\mu$L of DMSO were mixed with sterile water to reach the desired soil water potential in each soil. We mixed the liquid with the artificial soils and sealed the vials using a hydrophobic porous sealing film and incubated them for 1 h at 30°C in a static incubator. We grew *Ec-AHL-MHT* to exponential phase (OD$_{600}$ = 0.5) in LB at 37°C, washed the cells twice, and resuspended them in concentrated (2×) MIDV1 and 0.25M MOPS, pH 7.0. After AHL incubation in the soils, vials were opened and 10$^8$ cells in 100 $\mu$L of medium were added. Vials were capped every 3 min and incubated at 30°C without shaking. After 6 h, gas

**TABLE 3** Composition of the Austin series artificial soils[a]

| Soil name | Sand (%) | Silt (%) | Clay (%) | Mineralogy (%) (montmorillonite) | CaCO3 (%) | Xanthan (%) |
|---|---|---|---|---|---|---|
| PS | 13.3 | 32.1 | 54.7 | | | |
| PS+M | 13.3 | 32.1 | | 54.7 | | |
| PS+M+pH | 13.2 | 31.8 | | 54.2 | 1.0 | |
| PS+M+pH+OM | 13.0 | 31.4 | | 53.6 | 1.0 | 1.1 |

[a]Percentage of sand, silt, and clay quartz-based soil particles to mimic particle size (PS). Clay particles are replaced by montmorillonite to simulate mineralogy (M). pH is adjusted using $CaCO_3$. Xanthan was used as an OM source.

production was measured using GC-MS. The *Ec-AHL-MHT* signal was plotted as the ratio of the total mass of $CH_3Br$ normalized to $CO_2$.

**Mollisol experiments.** We used soil samples that had been previously collected from the A horizon of an Austin Series Mollisol (an Udorthentic Haplustoll) from a USDA facility near Temple, TX in 2009 (102). All soil samples were sieved through a 2 mm USA standard sieve and dried at 60°C, then stored. We measured C, H, and N by catalytic combustion and subsequent chromatographic separation and detection of $CO_2$, $H_2O$, and $N_2$ gases using a Costech ECS 4010 instrument. For OC measurement, inorganic C was removed first by full strength (7 M) HCl acid fumigation treatment in open-top silver capsules, following the procedure of Harris et al. (103). We measured particle size distribution using chemical dispersion followed by gravity sedimentation (104). Briefly, 30 g of soils were suspended in 3% hexametaphosphate (HMP) solution in a 250 mL HDPE bottle with 3:1 HMP (90 mL) to soil (30 g) ratio. The suspension was mixed on a reciprocating shaker for 2 h. The sand fraction was then collected using a 53 $\mu$m USA standard sieve; whereas the remaining silt and clay suspension fraction was stirred thoroughly and allowed to settle undisturbed in a 200 mL volumetric cylinder for a sedimentation period of 90 min. After the sedimentation, the suspended clay fraction was decanted. The sand fraction collected from sieving and the settled silt fraction was then dried in Al boats at 60°C to constant weight. The clay fraction was calculated by subtracting the weight of sand and silt from the original sample mass. Using the obtained information regarding particle size, OC, and pH of the natural soil, we created three artificial soils to mimic the soil properties with increased level of complexity (Table 3).

We tested the bioavailability of AHL in the Mollisol and artificial soils designed to mimic the properties of the Austin soil (Table 1). We built a suite of artificial soils of increasing complexity, mimicking particle size distribution (PS), then added minerals (PS+M) and adjusted pH (PS+M+pH). We fixed $\theta = 0.4$ g($H_2O$) g(soil)$^{-1}$. We picked and grew three MG1655-*las* colonies for 18 h while shaking at 37°C in LB. The next day, we made a 100× dilution of these cells and grew them to an $OD_{600}$ of 0.5 in LB at 37°C. We washed the cells twice. MIDV1 media was used for all experiments except when using the soil PS+M where MIDV1+0.25M MOPS was used to neutralize alkalinity of the minerals. *Ec-AHL-MHT* ($10^8$ cells) and 10× dilutions of 3-oxo-C12-HSL were mixed and immediately transferred into 2 mL glass vials containing the different artificial soils and the natural soil (2× autoclaved). A vial containing only the nutrients (no matrix) was included as a control. We capped the vials every 3 min, and gas was accumulated at 30°C in static for 6 h. Gas was measured using a GC-MS. Raw data was treated as described in Section 1.11.16. We report the *Ec-AHL-MHT* signal as the ratio of the total mass of $CH_3Br$ normalized by $CO_2$.

**Gas partitioning curves.** In all our experiments the $CH_3Br$ generated by biosensors in response to AHL needed to pass through a hydrated soil matrix before being measured in the vial headspace. This means that the signal was likely influenced by a number of sorption, dissolution, and diffusion processes. To account for this, we generated standard $CH_3Br$ curves in each artificial soil and hydration condition used in this study (Fig. S5). Standard curves were built for $CH_3Br$ using a 4× dilution in methanol of an analytical standard (2,000 $\mu$g/mL in methanol; Restek). We added 10× serial dilutions of the stock solution into 200 $\mu$L of MIDV1 media. We combined the dilutions with water needed to reach the desired hydration condition in the artificial soils, immediately mixed the liquid with the soils weighed into 2 mL vials, and rapidly crimped the vials to avoid gas loss. We incubated all vials for 6 h at 30°C in a static incubator to allow each gas to reach equilibration between the different phases prior to the GC-MS analysis.

We performed standard curves of $CO_2$ by reacting $H_2CO_3$ and $H_3PO_4$ to produce $CO_2$, $H_2O$, and $Na_3PO_4$. We added sterile 100 $\mu$L of 85% $H_3PO_4$ and 10× serial dilutions (in 100 $\mu$L) of a 200 mM $H_2CO_3$ stock solution to vials containing the different artificial soils and MIDV1 to achieve the desired $\theta$. The vials were immediately crimped, and gas was equilibrated for 6 h at 30°C in a static incubator. Gas partitioned into headspace was measured using a GC-MS. We fitted all curves of mass versus gas detected (peak area) to a log-log regression model using equation 6:

$$log(y) = m \times log(x) + b$$

**Gas counts per cells.** To determine how headspace gas concentrations and cell number relate, we grew biosensors to an $OD_{600}$ of 0.5 in LB. We washed the cells twice in MIDV1 and made three 10× serial dilutions starting from $10^5$ cells in 200 $\mu$L. Cells were added to 2 mL glass vials and incubated for 3 h at 30°C. After measuring gas production, we added LB (800 $\mu$L) to each vial and mixed the cultures for 3 min at 900 rpm. Serial dilutions of cells were spread on LB-agar plates using a glass cell spreader. After a

24-h incubation at 30°C, plates were imaged using a Nikon camera, and Image J 1.51 was used to count CFU. All gas signals were converted to mass, and plots of CFU versus gas were fit to equation 6.

**GC-MS analysis.** We measured $CO_2$ and $CH_3Br$ using an Agilent 7890B gas chromatograph and a 5977E mass spectrometer (GC-MS) using an Agilent 7693A liquid autosampler equipped with a 100 $\mu$L gastight syringe (Agilent G4513-80222). Headspace gas (50 $\mu$L) was injected into a DB-VRX capillary column (20 m, 0.18 mm inner diameter, and 1 $\mu$m film) at a 50:1 split ratio and the following oven temperature gradient was used to separate the gasses: an initial hold at 45°C for 84 s, then a transition to 60°C at 36°C per minute, and a final hold at 60°C for 10 s. MS analysis was performed using selected ion monitoring mode for $CO_2$ (MW = 44 and 45) and $CH_3Br$ (MW = 94 and 96). We used Agilent MassHunter Workstation Quantitative Analysis software to quantify the peak area of the major ions and used the minor ions as qualifiers.

**Statistics.** All data presented in this paper was processed by converting the raw $CH_3Br$ and $CO_2$ signal (peak area) into total mass of gas using the chemical standard curves. For experiments in which gas production was measured over time (cell survival and bioavailability assays), we calculated the mass removed per each injection and added the cumulative sum to obtain total mass per each time point. A minimum of three replicates was used in each experiment and all data is presented as the average values with error bars representing $\pm 1$ standard deviation. The data was processed in Python 3.7.1 and plotted using GraphPad Prism 8.

## SUPPLEMENTAL MATERIAL

Supplemental material is available online only.
**FIG S1**, TIF file, 2.3 MB.
**FIG S2**, TIF file, 0.7 MB.
**FIG S3**, TIF file, 0.8 MB.
**FIG S4**, TIF file, 0.3 MB.
**FIG S5**, TIF file, 1.4 MB.
**FIG S6**, TIF file, 2.6 MB.
**FIG S7**, TIF file, 1.6 MB.
**FIG S8**, TIF file, 0.5 MB.
**TABLE S1**, XLSX file, 0.01 MB.
**TABLE S2**, XLSX file, 0.01 MB.

## ACKNOWLEDGMENTS

We are grateful for financial support from the W.M. Keck Foundation (to C.A.M. and J.J.S.), Defense Advanced Research Projects Agency HR0011-19-2-0019 (to J.J.S. and C.A.M.), and William Marsh Rice University.

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
