## [Reviewer comments · mSystems]

Artificial soils reveal individual factor controls on microbial processes

Ilenne Del Valle, Xiaodong Gao, Teamrat Ghezzehei, Jonathan Silberg, and Caroline Masiello

Corresponding Author(s): Caroline Masiello, Rice University

Review Timeline:

Submission Date:	March 23, 2022
Editorial Decision:	April 29, 2022
Revision Received:	June 12, 2022
Accepted:	July 3, 2022

Editor: Nick Bouskill

Reviewer(s): Disclosure of reviewer identity is with reference to reviewer comments included in decision letter(s). The following individuals involved in review of your submission have agreed to reveal their identity: Michael Smanski (Reviewer #1); Kristen M DeAngelis (Reviewer #2)

Transaction Report:

DOI: <https://doi.org/10.1128/msystems.00301-22>

April 29, 2022

Dr. Caroline A Masiello
Rice University
Earth, Environmental, and Planetary Sciences
6100 Main Street, MS-126
Houston, TX 77005

Re: mSystems00301-22 (Artificial soils reveal individual factor controls on microbial processes)

Dear Dr. Caroline A Masiello:

Thank you for submitting your manuscript to mSystems. We have completed our review and I am pleased to inform you that, in principle, we expect to accept it for publication in mSystems. However, acceptance will not be final until you have adequately addressed the reviewer comments.

Preparing Revision Guidelines

Sincerely,

Nick Bouskill

Editor, mSystems

Journals Department
Reviewer comments:

Reviewer #1 (Comments for the Author):

Attached.

Reviewer #2 (Comments for the Author):

I am a huge fan of the artificial soil system as a way of understanding emergent properties of soil and soil microbes, so I was super excited to get to review this manuscript. While many artificial soil papers have been published, this approach is (for now) underutilized, perhaps, as the authors suggest, because we don't have good, tunable methods for generating them.

The authors goal is to present a tuneable artificial soil protocol that can be used to systematically explore how soil edaphic factors affect microbial dynamics, and on this front I'd say they were successful. Artificial soils were constructed over a range of particle sizes and OM contents, and then results used to conclude that "OM controls over microbial growth are complex." An important caveat of this nice work is that microbes are exogenously added to artificial soils. Please address how they might behave differently than those that are grown in and on soils in the discussion. For example, given a couple of generation times, microbes may be able to make an EPS matrix that would contribute to the soil OM content and make up for some of the water potential issues.

The authors main claim for this research is that additive versus emergent properties of microbial communities can be discerned using this approach. The support for this claim is less clearly articulated. It is addressed in the discussion based on the observation that adding more OM does not linearly affect microbial survival or growth. The authors acknowledge that organo-mineral interactions during OM formation are important (L524-538), but these do not have to be "decades to millennium scale" time frames. Other artificial soil studies have achieved this in 9 months (<https://doi.org/10.1038/s43705-021-00071-7>) or 15 months (<https://doi.org/10.1038/ncomms13630>). This paper is very long, so it is possible that this point was made, just not clearly. I think that the missing 'emergent' property is probably literally the emergence of EPS produced by endogenous microbes, which is largely missing from this otherwise very comprehensive recipe and protocol.

Well done! I am excited for this to be published so I can share it.

Minor comments

- The abstract is a bit misleading by referring to "Gram negative bacteria" and "Gram positive bacteria" when one strain of each was used. I suggest just naming the strains.
- Please explain the idea behind the methyl bromide in the methods (I read the methods before results & discussion, since that's more my style and that's how it would probably be published in Spectrum).
- Why is it important for the peds to survive autoclaving (L186)? If artificial soils are allowed to 'grow' naturally, this step is not necessary towards creating gnotobiotic environments e.g. <https://doi.org/10.1038/s41467-020-17502-z>
- Can you relate gas rate of production to cell turnover time, or estimated turnover time? (L275-281)

Review for: “Artificial soils reveal individual factor controls on microbial processes”

Authors: Del Valle I, Gao X, Ghezzehei TA, Silberg J, Masiello C

Reviewed by: Michael Smanski, smanski@umn.edu

Summary:

The authors develop a model system for studying microbes in soil-like environments. The artificial soils are less complex than natural soils but capture many of the physical-chemical properties of soils better than traditional laboratory culture methods. They characterize several artificial soils, test how several soil variables impact the viability and signal processing of engineered bacterial strains, and show that behavior of an engineered *E. coli* strain in natural soil is more closely matched by their artificial soils than by liquid culture.

I am not an expert on the physicochemical characterization of soils, so hopefully another reviewer can comment on the first part of the manuscript. However, I have thought a lot about microbial signaling in soil environments. From that perspective, this is an exciting paper. The way we study microbial interactions in the lab currently are so far removed from the microenvironment present in soil, that any discoveries need to be taken with a lot of skepticism. The artificial soils presented by this group offer a great ‘next step’ to test hypotheses on microbial interactions and signaling in something that is closer to the natural environment.

My overall review of the paper is positive. I think that this will be well-received and others will be interested in using these artificial soils. There are a few weaknesses that, if addressed, would make the paper stronger and more compelling. Because the methylhalide reporter system is not likely to be widely adopted (too much specialized equipment is required), it would be great to see if the artificial soils lend themselves better to things like RNA/protein/metabolite extraction and analysis than do natural soils. This suggestion should not be seen as a barrier to publishing the paper in close to its current form, but as a good ‘next step’ for future work.

The paper is well-written and free of grammatical errors. The figures cleanly communicate the most important results. I did not look at the supplementary figures. Some weaknesses that should be addressed prior to publication include revising some language concerning system additivity and the application of Hill-equation fitting. These are described in more detail below.

In my opinion, the results in this paper warrant publication in mSystems provided the authors address the weaknesses noted below my signature. I welcome any questions or concerns regarding this review to be directed to me via email.

Best,

Michael Smanski
smanski@umn.edu

Weaknesses

1. The major critique that I have of the manuscript is the discussion of system additivity. Additive systems behave such that for two input/outputs (X_1/Y_1 and X_2/Y_2): The addition of both inputs (X_1+X_2) leads to an output which is the sum of their individual outputs (Y_1+Y_2). The experiment performed in Figure 6, which is described as showing additive behavior, was not performed in a way to test if the system has additive behavior. Variables would have to be tested in isolation (PS+M, PS+pH, PS+OM, etc) to determine if making two modifications to the soil (e.g. PS+M+pH) are additive or display positive or negative synergy. Another way to pose the question is: if complexity were increased in different orders (OM, OM+M, OM+M+PS; or PS, PS+OM, PS+OM+pH), would the same conclusions be reached (that PS is the primary driver of A and M is the primary driver of k)?
2. Statistical analysis of the bar graphs should be performed. Due to the variance in measurement, it is important to include a statistical test to determine which treatments produce (for example in Figure 6bc) effects that are different from each other in a statistically relevant way.
3. I could be wrong about this (e.g. the authors are welcome to rebut this claim if they think it is unfair), but I think the Hill equation should only be applied to data that have a clear sigmoidal curve. The M2a data in Figure 5a is still exponentially increasing (i.e. it has not reached the inflection point), so I would not trust the estimation of k that fitting a Hill equation provides. I think it would better to say that this treatment shifted the response curve outside of the range of testing, so the k value was changed by *at least* 5 orders of magnitude. Some of the data in Fig 6a are also getting close to the point where a Hill fit is questionable.

Minor Revisions:

1. P10L203, The figure citation should say Fig1b-d, as Fig1b is just pictures of vials and not the characterization referenced in that sentence.
2. P11L232, change to “while the surface area of soil containing montmorillonite (M2a) was two orders of magnitude greater”. (‘larger’ still works, but main need here is to make the subject of the clause ‘surface area’ and not ‘soil’).
3. P12,13: I am not confident in the assumption that the relationship between CH₃X production and cell growth (CFUs) that was determined in liquid culture can be assumed to hold true in the artificial soils. I think you would have to measure the correlation between gas production and isolated CFUs from the soil to make that claim. A safer and easier approach would just be to report what you observe, for example change P13L266 to “To evaluate how particle size distribution effects the gas reporter activity...”. I do not think this would decrease the impact of the manuscript, and it is easier to defend.
4. P13L276-286. This whole paragraph was difficult for me to digest. I think it might stem from your use of the word ‘signal’ in the first sentence. My current understanding (which is still not clear) is that you use ‘signal’ here to mean “a measurement that can be made using our experimental set-up”. Is thus true? If so, I recommend rephrasing, since so much of the rest of the paper is about microbial signaling, in which signal means “a small molecule used in bacterial communication”.

5. P18L380: See weaknesses section above. I think you should change “5 orders of magnitude” and say that k could not be measured reliably because it was shifted so far right.
6. P18L400: If my understanding is correct, this series of artificial soils are distinct from the series that was the subject of the first part of the paper. If so, I suggest adding the word ‘new’ (we generated a new series of artificial soils), otherwise readers (like me) will think you are making another claim about the original series of artificial soils.
7. P21L451: I do not think that the question of system additivity was rigorously tested in this paper (see above).
8. P22L473: While these OM sources are not metabolized by these bacteria, could they still have a biological impact by acting as signals? I mention this because it might still be more complicated than OM physicochemical effects on soils, as is claimed.
9. P24L513: note that the 5 orders of magnitude mentioned here should be revised based on comments above)
10. P24L523: olipeptide? I am not familiar with this word. Do you mean polypeptide, or oligopeptide?
11. P24L526: I disagree that the Mollisol experiment showed additive effects, because you do not know what the individual effects are.
12. P24L530: Is the claim that OM addition does not bring artificial soils closer to natural soils stem from Figure 6BC? If so, I think this is a poorly justified claim. In 6C, there is not opportunity to bring them closer to NS, because k is already there. In 6B, it is not clear that the experimental design had sufficient statistical power to even make a measurement of this effect size, given the variance in measurement.
13. P30L661: please define the composition of LB used in this study, as there are several varieties.
14. P35L771: Please remove the clause ‘until exponential phase’. In fact, they had been in exponential phase earlier, but have started exiting the physiological exponential phase by $OD_{600} = 0.3$ (<https://journals.asm.org/doi/full/10.1128/JB.01368-07>) (which is earlier than most people canonically believe).
15. Figure 2c: There is a claim in the manuscript that E coli peaks at 4 hours. Shouldn’t it be 8 hours?
16. Figure 2: Please be more precise about the normalization in this figure. Are data normalized across experiments (soil types) or only within a soil type?
17. Figure 6BC, what is the height of the bar supposed to represent? I would assume it represents the mean value from three replicates, but (particularly for the yellow bar) this does not seem to be the case. My naïve guess is that it is plotting the mean of two replicates, but the third replicate was not included when making the bar graphs (could be a simple excel error, but you should double check).

Dear Dr. Bouskill,

We thank the referees for their feedback on our manuscript (**mSystems00301-22**). We refined the manuscript to address their comments and provide a point-by-point description of our corrections below. Text changes in our manuscript are highlighted in red and a cleared draft is provided. Additionally, the page numbers below refer to the marked up manuscript. We hope that our manuscript is now acceptable for publication.

REVIEWER 1 COMMENTS

1. Summary comments. Reviewer #1 noted: “The authors develop a model system for studying microbes in soil-like environments. The artificial soils are less complex than natural soils but capture many of the physical-chemical properties of soils better than traditional laboratory culture methods. They characterize several artificial soils, test how several soil variables impact the viability and signal processing of engineered bacterial strains, and show that behavior of an engineered *E. coli* strain in natural soil is more closely matched by their artificial soils than by liquid culture. I am not an expert on the physicochemical characterization of soils, so hopefully another reviewer can comment on the first part of the manuscript. However, I have thought a lot about microbial signaling in soil environments. From that perspective, this is an exciting paper. The way we study microbial interactions in the lab currently are so far removed from the microenvironment present in soil, that any discoveries need to be taken with a lot of skepticism. The artificial soils presented by this group offer a great ‘next step’ to test hypotheses on microbial interactions and signaling in something that is closer to the natural environment. My overall review of the paper is positive. I think that this will be well-received and others will be interested in using these artificial soils. There are a few weaknesses that, if addressed, would make the paper stronger and more compelling. Because the methylhalide reporter system is not likely to be widely adopted (too much specialized equipment is required), it would be great to see if the artificial soils lend themselves better to things like RNA/protein/metabolite extraction and analysis than do natural soils. This suggestion should not be seen as a barrier to publishing the paper in close to its current form, but as a good ‘next step’ for future work. The paper is well-written and free of grammatical errors. The figures cleanly communicate the most important results. I did not look at the supplementary figures. Some weaknesses that should be addressed prior to publication include revising some language concerning system additivity and the application of Hill-equation fitting. These are described in more detail below. In my opinion, the results in this paper warrant publication in *mSystems* provided the authors address the weaknesses noted below my signature. I welcome any questions or concerns regarding this review to be directed to me via email.”

We thank this reviewer for these positive comments. We agree that these soils represent a nice model system for performing -omics and are working to apply them for such studies with model soil consortia (see <https://doi.org/10.3389/fmicb.2020.01987>).

2. Clarify the description of additivity. Reviewer #1 noted: “The major critique that I have of the manuscript is the discussion of system additivity. Additive systems behave such that for two input/outputs (X_1/Y_1 and X_2/Y_2): The addition of both inputs (X_1+X_2) leads to an output which is the sum of their individual outputs (Y_1+Y_2). The experiment performed in Figure 6, which is described as showing additive behavior, was not performed in a way to test if the system has additive behavior. Variables would have to be tested in isolation (PS+M, PS+pH, PS+OM, etc) to determine if making two modifications to the soil (e.g. PS+M+pH) are additive or display positive or negative synergy. Another way to pose the question is: if complexity were increased in different orders (OM, OM+M, OM+M+PS; or PS, PS+OM, PS+OM+pH), would the same conclusions be reached (that PS is the primary driver of A and M is the primary driver of k)?”

This is a great point. We deleted the last sentence of the Abstract to avoid overreaching on our conclusions, as well as a sentence from the Importance paragraph. To clarify our results, we adjusted the text on Pages 25-26 to reflect our findings from statistical analysis performed in response to Point #3 below. Specifically, we adjusted the text in the Discussion to note:

*“Our measurements comparing AHL bioavailability in a Mollisol and within artificial soils that mimic different levels of complexity within the Mollisol revealed that **the primary factor affecting maximum gas production is soil properties have additive effects, including particle size distribution. In addition, the AHL concentration needed for half-maximum induction of the biosensor output was attenuated to the greatest by the addition of soil mineralogy and pH. In the case where** These results illustrate how artificial soils **reveal additive properties, artificial soils can** will be useful in future studies to give mechanistic information on how different matrix properties affect biological activity. As such, artificial soils will be useful for informing predictive models to better understand natural soils' influence on biological processes. **Artificial soils also carry information when they fail to behave additively, as they help us understand which soil properties are emergent.**”*

3. Include additional statistical analysis. Reviewer #1 noted: “Statistical analysis of the bar graphs should be performed. Due to the variance in measurement, it is important to include a statistical test to determine which treatments produce (for example in Figure 6bc) effects that are different from each other in a statistically relevant way.”

We appreciate this comment. We added more statistical analysis (see Figure 6), which revealed that all artificial soils present a significant change in maximum gas production compared with the measurements in liquid culture ($P < 0.001$). Setting mineralogy and particle size to match the original Mollisol did not lead to a significant difference compared to setting particle size alone. However, setting of mineralogy, particle size, and pH led to a significant difference from setting particle size alone ($P < 0.03$). Inclusion of organic matter along with setting mineralogy, particle size, and pH was not significantly different from setting particle size alone, particle size and mineralogy, or particle size, mineralogy, and pH. Soil mimicking all properties studied is significantly different from natural soil ($P < 0.03$). The bioavailability of the AHL cell-cell signal (k) was statistically significantly different between the liquid control and artificial soils mimicking mineralogy and pH. We have added the following text to note these differences in Page 20:

*“The synthetic soil series gives us insight into the parameters driving this change: comparing the synthetic soil series A values to those generated by biosensor growth in liquid media, we found the largest change in AHL response was driven simply by the addition of a physical matrix (quartz particles), mimicking only the particle size distribution of the natural soil ($P < 0.001$). In the quartz-only artificial soil the maximum gas production (A) decreased by half compared to liquid media. These results underline the importance of the physical matrix in controlling microbial accessibility of diffusible compounds. **Adding mineralogy and pH (PS+M+pH) to match the Mollisol content also significantly changed the maximal gas production from particle size (PS) alone ($P < 0.03$), although particle size and mineralogy (PS+M) were not significantly different from PS alone. Finally, soil containing organic matter (PS+M+pH+OM) was not significantly different from PS+M+pH, PS+M, or PS. However, it was significantly different from the Mollisol ($P < 0.03$).**”*

We also modified the text on Page 21:

“The k values observed in liquid culture and PS+M+pH experiments were significantly different ($p < 0.03$).”

Furthermore, we refined the legend for Figure 6 to reflect this new statistical analysis:

“(B) Maximum gas production ($\text{CH}_3\text{Br}/\text{CO}_2$) obtained from a fit of the data to the Hill equation reveals a decrease in maximum gas production as soil complexity increases (Ordinary one-

way ANOVA, Tukey's multiple comparisons, ** $p < 0.001$, * $p < 0.03$). (C) Amount of AHL (pM) necessary for a half maximum gas response. Artificial soil that recreates texture, mineralogy, and pH requires an AHL concentration within the same order of magnitude as the natural soil to induce the biosensor (Non-parametric ANOVA, Dunn's multiple comparisons, * $p < 0.03$). Error bar represents one standard deviation calculated from three replicates."

4. Clarify the description of trends. Reviewer #1 noted: "I could be wrong about this (e.g. the authors are welcome to rebut this claim if they think it is unfair), but I think the Hill equation should only be applied to data that have a clear sigmoidal curve. The M2a data in Figure 5a is still exponentially increasing (i.e. it has not reached the inflection point), so I would not trust the estimation of k that fitting a Hill equation provides. I think it would better to say that this treatment shifted the response curve outside of the range of testing, so the k value was changed by at least 5 orders of magnitude. Some of the data in Fig 6a are also getting close to the point where a Hill fit is questionable."

We have refined the text as suggested on Page 18. This text now states:

"We found that soil containing 2:1 clays increased the AHL concentration needed to achieve a half maximum indicator gas production (k) by up to 3 orders of magnitude for a non-expanding clay (I2a) and shifted the response curve outside of the range of testing, indicating that the k value was changed by at least 5 orders of magnitude for an expanding clay (M2a) compared with soils made of quartz (Q2a)."

We also refined the legend in Figure 5 as follows:

"Figure 5 legend: "different k values were obtained for liquid (7.8×10^{-11}), Q2a (3.1×10^{-11}), K2a (2.8×10^{-11}), I2a (2.9×10^{-8}), and M2a ($\geq 4.2 \times 10^{-6}$)..."

Furthermore, we adjusted our text on Page 20 to note:

"We found that the liquid control and the artificial soil that only resembles the natural soil's particle size (PS) have k values that differ by <2.5-fold ~~within the same order of magnitude~~ (~110 pM and ~244 pM respectively), while adding ~~mineralogy~~ mineralogical complexity (PS+M) increases k by at least two orders of magnitude (~39 nM). This result suggests that sorption of the AHL onto the ~~minerals~~ mineral surfaces in the Mollisol decreases AHL bioavailability."

5. Refine results text. Reviewer #1 noted: "P10L203, The figure citation should say Fig1b-d, as Fig1b is just pictures of vials and not the characterization referenced in that sentence." and suggested: "P11L232, change to "while the surface area of soil containing montmorillonite (M2a) was two orders of magnitude greater". ('larger' still works, but main need here is to make the subject of the clause 'surface area' and not 'soil')."

We have refined the text as suggested, which now states:

"To determine how artificial soil composition relates to physicochemical properties, we characterized the water retention, surface area, and pH of each soil (Fig. 1b-d)."

We also changed the text on Page 11 so that it now notes:

"while the soil containing montmorillonite (M2a) was two orders of magnitude greater than Q2a large."

6. Clarify rationale. Reviewer #1 noted: "P12,13: I am not confident in the assumption that the relationship between CH₃X production and cell growth (CFUs) that was determined in liquid culture can be assumed to hold true in the artificial soils. I think you would have to measure the correlation between gas production and isolated CFUs from the soil to make that claim. A safer and easier approach would just be to report what you observe, for example change P13L266 to "To evaluate how particle size distribution effects the gas reporter activity...". I do not think this would decrease the impact of the manuscript, and it is easier to defend."

This is a great point. We have refined the text as suggested. The text now states:

“To evaluate how particle size distribution affects ~~cell-growth-gas reporter activity~~, we added our gas-reporting microbes (10^6 cells) to three soils (Q1, Q2a, Q3a - 800 mg each) held at a constant water content (θ) of 0.25 g(H_2O)/g(soil). At this θ , the soil matric potential (ψ_m) varies across soils.”

7. Refine paragraph so more accessible. Reviewer #1 noted: “P13L276-286. This whole paragraph was difficult for me to digest. I think it might stem from your use of the word ‘signal’ in the first sentence. My current understanding (which is still not clear) is that you use ‘signal’ here to mean “a measurement that can be made using our experimental set-up”. Is thus true? If so, I recommend rephrasing, since so much of the rest of the paper is about microbial signaling, in which signal means “a small molecule used in bacterial communication”.”

We refined the text as suggested. The text now states:

*“~~Cell growth slows as nutrients are consumed, so we~~ We expected indicator gas production to decrease with time ~~as cell growth slows down due to nutrient consumption potentially creating an additional signal in our particle size experiments~~. To test this ~~idea~~, we monitored the rate of CH_3Br production in soils having different particle sizes (**Fig S6a**)...”*

8. Adjust description of result. Reviewer #1 noted: “P18L380: See weaknesses section above. I think you should change “5 orders of magnitude” and say that k could not be measured reliably because it was shifted so far right.”

We refined the text as suggested on Page 18. This text now states:

“We found that soil containing 2:1 clays increased the AHL concentration needed to achieve a half maximum indicator gas production (k) by up to 3 orders of magnitude for a non-expanding clay (I2a) and ~~shifted the response curve outside of the range of testing, indicating that the k value was changed by at least 5 orders of magnitude for an expanding clay (M2a) compared with soils made of quartz (Q2a).~~”

9. Add a word. Reviewer #1 noted: “P18L400: If my understanding is correct, this series of artificial soils are distinct from the series that was the subject of the first part of the paper. If so, I suggest adding the word ‘new’ (we generated a new series of artificial soils), otherwise readers (like me) will think you are making another claim about the original series of artificial soils.”

We have added a word as suggested:

*“we generated a **new** series of artificial soils that sequentially recreated the properties of a natural soil”*

10. Clarify additivity description. Reviewer #1 noted: “P21L451: I do not think that the question of system additivity was rigorously tested in this paper (see above).”

We have refined the manuscript to be more precise in describing our experiments and avoid overreaching on conclusions. The text on this page now notes:

“We additionally tested the generality of one underlying assumption of the artificial soil design: that ~~some soil properties present are~~ additive behaviors ~~when starting with texture as the simplest component evaluated, not emergent~~. To track microbial response we used biosensors: microbes that report on their environment and/or their behaviors.”

11. Question about organic matter source. Reviewer #1 noted: “P22L473: While these OM sources are not metabolized by these bacteria, could they still have a biological impact by acting as signals? I mention this because it might still be more complicated than OM physicochemical effects on soils, as is claimed.”

This is an interesting point. In the environment, organic matter could contain diverse signals that impact cell behavior. However, the OM used in our study is homogeneous and cannot be

metabolized. It is possible that cells could physically bind to this OM, and this interaction could lead to changes in cellular gene expression. However, we did not find any evidence in the literature that pure xanthan or chitin act as a signal to extracellular sensor components (e.g., two component systems that microbes use for sensing) in *E. coli* MG1655 or *B. subtilis* PY79.

12. Adjust description of results. Reviewer #1 noted: “P24L513: note that the 5 orders of magnitude mentioned here should be revised based on comments above)”

We refined the manuscript as suggested:

“Our artificial soil measurements showed that mineralogy could decrease the AHL concentration by ~~at least up to~~ five orders of magnitude...”

13. Change a word. Reviewer #1 noted: “P24L523: olipeptide? I am not familiar with this word. Do you mean polypeptide, or oligopeptide?”

We thank the reviewer for finding this typo, which we have corrected. The text now reads:

*“In future studies, it will be interesting to sample soils with an even more comprehensive range of pH and mineralogy properties and determine if these physicochemical properties are also essential factors impacting other microbial signals' persistence, such as short chain AHLs, and ~~olipeptide~~ **peptide** autoinducers.”*

14. Clarify description of additivity. Reviewer #1 noted: “P24L526: I disagree that the Mollisol experiment showed additive effects, because you do not know what the individual effects are.”

As noted in our response to prior comments above, we refined the manuscript text to be more precise in our conclusions and avoid overreaching. The text on this page now notes:

“Our measurements comparing AHL bioavailability in a Mollisol and within artificial soils that mimic different levels of complexity within the Mollisol revealed that ~~the primary factor affecting maximum gas production is soil properties have additive effects, including~~ particle size distribution. In addition, the AHL concentration needed for half-maximum induction of the biosensor output was attenuated to the greatest by the addition of soil mineralogy and pH. ~~In the case where~~ These results illustrate how artificial soils ~~reveal additive properties, artificial soils can~~ will be useful in future studies to give mechanistic information on how different matrix properties affect biological activity. As such, artificial soils will be useful for informing predictive models to better understand natural soils' influence on biological processes. ~~Artificial soils also carry information when they fail to behave additively, as they help us understand which soil properties are emergent.~~”

15. Better defend claims. Reviewer #1 noted: “P24L530: Is the claim that OM addition does not bring artificial soils closer to natural soils stem from Figure 6BC? If so, I think this is a poorly justified claim. In 6C, there is not opportunity to bring them closer to NS, because k is already there. In 6B, it is not clear that the experimental design had sufficient statistical power to even make a measurement of this effect size, given the variance in measurement.”

This is a great point. As noted in our responses to Points #2, 3, and 14 above, we refined our discussion to be more precise in our description of the results.

16. Define composition of growth medium. Reviewer #1 noted: “P30L661: please define the composition of LB used in this study, as there are several varieties.”

We included the composition as suggested. This text now states:

*“Lysogeny Broth (LB) **containing 10 g/L tryptone, 5 g/L yeast extract, and 10 g/L sodium chloride** was used for culturing strains and all engineering.”*

17. Remove a clause. Reviewer #1 noted: “P35L771: Please remove the clause ‘until exponential phase’. In fact, they had been in exponential phase earlier, but have started exiting the physiological exponential phase by OD600 = 0.3

(<https://journals.asm.org/doi/full/10.1128/JB.01368-07>) (which is earlier than most people canonically believe).”

We removed the text as suggested. The text now states:

*“The next day, we made a 100x dilution of these cells and grew them **until exponential phase (to an OD600 of 0.5)** in LB at 37°C.”*

18. Adjust data description. Reviewer #1 noted: “Figure 2c: There is a claim in the manuscript that E coli peaks at 4 hours. Shouldn't it be 8 hours?”

We thank the reviewer for catching this typo. The text has been refined as follows:

*“In the sandy artificial soil, Ec-MHT and Bs-MHT reached a maximum cell density at 20 and **4 hours 8 hours**, respectively (**Fig. 2c**).”*

19. Clarify description of normalization. Reviewer #1 noted: “Figure 2: Please be more precise about the normalization in this figure. Are data normalized across experiments (soil types) or only within a soil type?”

We expanded our description about normalization on Page 33. This text now states:

*“To allow comparison between **experiments strains**, CH₃Br (µg/mL) was normalized to the maximum signal observed **across different soil types**.”*

20. Clarify amplitude of data. Reviewer #1 noted: “Figure 6BC, what is the height of the bar supposed to represent? I would assume it represents the mean value from three replicates, but (particularly for the yellow bar) this does not seem to be the case. My naïve guess is that it is plotting the mean of two replicates, but the third replicate was not included when making the bar graphs (could be a simple excel error, but you should double check).”

We initially plotted the median rather than the mean for the bar graphs in this figure. We refined this figure so that it now shows the mean.

REVIEWER 2 COMMENTS

1. Summary comments. Reviewer #2 noted: “I am a huge fan of the artificial soil system as a way of understanding emergent properties of soil and soil microbes, so I was super excited to get to review this manuscript. While many artificial soil papers have been published, this approach is (for now) underutilized, perhaps, as the authors suggest, because we don't have good, tunable methods for generating them.”

We thank this reviewer for these positive comments and for noting “Well done! I am excited for this to be published so I can share it.” We really appreciate the positive comments and thoughtful feedback on our study.

2. Discuss how introduction of microbes affects growth in soils. Reviewer #2 stated: “The authors goal is to present a tuneable artificial soil protocol that can be used to systematically explore how soil edaphic factors affect microbial dynamics, and on this front I'd say they were successful. Artificial soils were constructed over a range of particle sizes and OM contents, and then results used to conclude that “OM controls over microbial growth are complex.” An important caveat of this nice work is that microbes are exogenously added to artificial soils. Please address how they might behave differently than those that are grown in and on soils in the discussion. For example, given a couple of generation times, microbes may be able to make an EPS matrix that would contribute to the soil OM content and make up for some of the water potential issues.”

This is a fabulous point. When using artificial soils, microbes are added in a planktonic form, yet *in situ* they partition into both planktonic and biofilm forms. As such the initial trends observed with soils do not reflect a soil that contains significant amounts of exopolysaccharide biofilms containing microbes. To mimic complex soil dynamics, longer duration incubations will be

required with soil consortia, where the microbes are given time to consume the initial nutrients and form biofilms. Additionally, our studies used individual microbes in the artificial soils, which are not typically found in isolation in the environment. In situ, soils contain consortia, which dynamically consume and store carbon through a range of processes, leading to heterogeneity within soil aggregates. In some locations, microbes catabolize organic matter leading to a net loss of carbon, while biofilm and necromass formation in other locations can lead to organic matter storage. To better place our results in this context, we have added the following paragraph on Page 23 that describes these very important considerations.

“Biosensors could also be incubated in the presence of microbial communities capable of metabolizing different OM sources to evaluate differences in water and nutrient availability simultaneously (77). A complementary approach would be to incorporate microbial communities for longer incubations, allowing them to grow and form biofilms, and naturally microaggregate the synthetic matrix, thereby dynamically changing the soil OM content. Biosensors could be incorporated at different time points to report changes in water retention properties. A potential limitation for the latter approach is the large number of biosensor cells (10^6 to 10^8) required for a detectable gas signal, which could artificially shift the abundance of microbial community members.”

3. Clarify conclusion. The authors main claim for this research is that additive versus emergent properties of microbial communities can be discerned using this approach. The support for this claim is less clearly articulated. It is addressed in the discussion based on the observation that adding more OM does not linearly affect microbial survival or growth. The authors acknowledge that organo-mineral interactions during OM formation are important (L524-538), but these do not have to be "decades to millennium scale" time frames. Other artificial soil studies have achieved this in 9 months (<https://doi.org/10.1038/s43705-021-00071-7>) or 15 months (<https://doi.org/10.1038/ncomms13630>). This paper is very long, so it is possible that this point was made, just not clearly. I think that the missing 'emergent' property is probably literally the emergence of EPS produced by endogenous microbes, which is largely missing from this otherwise very comprehensive recipe and protocol.

We recognize that this manuscript is long, as we were working to be thorough to show how individual soil properties affect microbes to illustrate the power of the synthetic matrices. Over the short time course of our experiments, which are distinct from those noted, we do not expect that the microbes are dramatically altering the organic matter content of the soils. Instead, we found that organic matter attenuates the effects of other soil matrix properties on the bioavailability of the cell-cell signal. As we increased the complexity of soils incrementally, we found that addition of texture, mineralogy, and pH decreased the bioavailability of the signal more and more as each component was included in the soil. However, addition of organic matter led to higher bioavailability than this mixture, suggesting that it was interacting with other components (or the microbes) in a way that increased bioavailability. To clarify our results, which was requested by Reviewer #1, we have adjusted the text in on Pages 25-26 to more clearly convey our results as well as the importance of additional experiments in future studies:

“Our measurements comparing AHL bioavailability in a Mollisol and within artificial soils that mimic different levels of complexity within the Mollisol revealed that ~~the primary factor affecting maximum gas production is soil properties have additive effects, including~~ particle size distribution. In addition, the AHL concentration needed for half-maximum induction of the biosensor output was attenuated to the greatest by the addition of soil mineralogy and pH. ~~In the case where~~ These results illustrate how artificial soils ~~reveal additive properties, artificial soils can~~ will be useful in future studies to give mechanistic information on how different matrix properties affect biological activity. As such, artificial soils will be useful for informing predictive models to better understand natural soils' influence on biological processes. ~~Artificial soils~~

~~also carry information when they fail to behave additively, as they help us understand which soil properties are emergent.”~~

We also deleted a sentence from the Abstract and Importance paragraphs to more precisely convey our results.

4. Name strains in abstract. Reviewer #2 suggested: “The abstract is a bit misleading by referring to “Gram negative bacteria” and “Gram positive bacteria” when one strain of each was used. I suggest just naming the strains.”

We changed the Abstract as follows and made similar changes to the Importance statement.

*“When we used standardized matrices with varying textures to culture gas-reporting biosensors, we found that a Gram-negative bacterium (*Escherichia coli*) grew best in synthetic silt soils, remaining active over a wide range of soil matric potentials, while a Gram-positive bacterium (*Bacillus subtilis*) preferred sandy soils, sporulating at low water potentials.”*

5. Add gas reporter description to the methods. Reviewer #2 noted: “Please explain the idea behind the methyl bromide in the methods (I read the methods before results & discussion, since that's more my style and that's how it would probably be published in Spectrum).”

We appreciate this suggestion as we hadn't considered the order previously. We added the following text to the Methods on Page 32 to provide more details:

*“These strains both represent indicator gas reporters of microbial growth and metabolism, since their synthesis of methyl halides can be used to track their relative growth in a variety of hard-to-image environments. To study the bioavailability of AHLs, we used a previously described *E. coli* MG1655-*las* biosensor (designated Ec-AHL-MHT here) that reports on the presence of 3-oxo-C₁₂-HSL by synthesizing CH₃Br (28). With this latter strain, methyl halide production is dependent upon detection of 3-oxo-C₁₂-HSL, and normalization of this gas signal to respiration allows for a per cell signal that can be related to the HSL detected by the cell population.”*

6. Explain rationale for autoclaving protocol. Reviewer #2 asked: “Why is it important for the peds to survive autoclaving (L186)? If artificial soils are allowed to 'grow' naturally, this step is not necessary towards creating gnotobiotic environments e.g. <https://doi.org/10.1038/s41467-020-17502-z>”

For the short time scales of our experiments, we sought to create soil aggregation as part of our recipe to mimic the spatial structure found in native soils. In the environment, such aggregation occurs over longer time scales, through abiotic and biotic processes, both of which we mimic in this study. The rationale is now noted on Page 9, which includes a new citation.

“We explored two methods to simulate natural soil structure. In the first approach, we used wet-dry cycling to aggregate quartz particles through weak adhesion (12). With this approach, mineral addition leads to increased aggregate stability. Although these inorganic peds are fragile, they can survive autoclaving, providing structure in the absence of organic carbon. With artificial soils, it is beneficial to have the option to create aggregates lacking OM so that individual soil properties can be varied independently. In the second method, we added extracellular polymeric substances (EPS) prior to subjecting the matrix to wet/dry cycles (Fig. S1d). In this study we added xanthan and chitin to soils at 0.5 and 1% OM (w/w). This approach was used rather than allowing microbes to grow and form EPS in the matrix, since this latter approach takes months to generate EPS while our approach generates OM in a single day (49).”

7. Question about cell turnover. Reviewer #2 asked: “Can you relate gas rate of production to cell turnover time, or estimated turnover time? (L275-281)”

This is an interesting question. We have not yet established a way to estimate cell turnover time using gas production.

MINOR TYPOGRAPHICAL CORRECTIONS

In addition, a final proofread of the document led to a few minor typographical corrections, listed below:

P. 5 “These artificial soils have a range of tunable features that can be independently varied, including particle size distribution, mineral composition, pH, and organic matter (OM) **content and composition.**”

The words “content and composition” were inserted.

P. 5: “By combining the artificial soils with a new synthetic biology **reporting** tool, gas biosensors (27, 28),”

The word “reporting” was inserted.

P. 7: “This step is critical because grain size regulates soil microbial diversity by providing different microenvironments (pore spaces) and **controlling** the diffusivity of nutrients (34).”

The word “controlling” was inserted.

P. 7: “The distribution among the three particles **sizes**”

The word “sizes” was added.

P. 8: “The quartz matrices used here (Q1, Q2a, and Q3a) are **at** a neutral pH,”

The word “at” was added.

P. 17: “We hypothesized that differences in mineral surface area will impact the amount of AHL **being sorb sorbed** into the matrices”

The word “sorb” was changed to “sorbed” and “being” was deleted.

P. 18: “Surprisingly, addition of a 1:1 clay (K2a) did not affect k, even though the K2a surface area is ~10-**fold** higher than Q2a.”

The letter “x” was added.

P. 20: “First, comparing A from biosensors grown **in** the natural Mollisol to A generated by biosensors grown in liquid media,”

The word “in” was added.

P. 20: “while adding **mineralogy mineralogical complexity**”

“Mineralogy” was changed to “mineralogical complexity.”

P. 21: “This result suggests that sorption **of the AHL** onto the **minerals mineral surfaces** in the Mollisol decreases AHL bioavailability.

“Of AHL” and “surfaces” were added.

P. 25: “Furthermore, alkaline artificial soil decreased AHL **bioavailable bioavailability** by three orders of magnitude.”

“Bioavailable” was changed to “bioavailability.”

July 3, 2022

Dr. Caroline A Masiello
Rice University
Earth, Environmental, and Planetary Sciences
6100 Main Street, MS-126
Houston, TX 77005

Re: mSystems00301-22R1 (Artificial soils reveal individual factor controls on microbial processes)

Dear Dr. Caroline A Masiello:

Your manuscript has been accepted, and I am forwarding it to the ASM Journals Department for publication. For your reference, ASM Journals' address is given below. Before it can be scheduled for publication, your manuscript will be checked by the mSystems production staff to make sure that all elements meet the technical requirements for publication. They will contact you if anything needs to be revised before copyediting and production can begin. Otherwise, you will be notified when your proofs are ready to be viewed.

Publication Fees:

If you would like to submit a potential Featured Image, please email a file and a short legend to mSystems@asmusa.org. Please note that we can only consider images that (i) the authors created or own and (ii) have not been previously published. By submitting, you agree that the image can be used under the same terms as the published article. File requirements: square dimensions (4" x 4"), 300 dpi resolution, RGB colorspace, TIF file format.

We recognize that the video files can become quite large, and so to avoid quality loss ASM suggests sending the video file via <https://www.wetransfer.com/>. When you have a final version of the video and the still ready to share, please send it to mSystems staff at mSystems@asmusa.org.

Sincerely,

Nick Bouskill
Editor, mSystems

Journals Department
Fig. S6: Accept
Fig. S1: Accept
Fig. S3: Accept
Table S2: Accept
Fig. S5: Accept
Fig. S7: Accept
Fig. S8: Accept
Fig. S2: Accept
Fig. S4: Accept
Table S1: Accept